



# Can a regional-scale reduction of atmospheric $CO_2$ during the COVID-19 pandemic be detected from space? A case study for East China using satellite $XCO_2$ retrievals

Michael Buchwitz[1], Maximilian Reuter[1], Stefan Noël[1], Klaus Bramstedt[1], Oliver Schneising[1], Michael Hilker[1], Blanca Fuentes Andrade[1], Heinrich Bovensmann[1], John P. Burrows[1], Antonio Di Noia[2,3], Hartmut Boesch[2,3], Lianghai Wu[4], Jochen Landgraf[4], Ilse Aben[4], Christian Retscher[5], Christopher W. O'Dell[6], David Crisp[7]

[1]Institute of Environmental Physics (IUP), University of Bremen, 28334 Bremen, Germany

[2]Earth Observation Science, University of Leicester, LE1 7RH, Leicester, UK

[3]NERC National Centre for Earth Observation, LE1 7RH, Leicester, UK

[4]SRON Netherlands Institute for Space Research, 3584 CA Utrecht, The Netherlands

[5]Directorate of Earth Observation Programmes, European Space Agency (ESA), ESRIN, 00044 Frascati, Italy

[6]Cooperative Institute for Research in the Atmosphere, Colorado State University (CSU), Fort Collins, CO 80523, USA

[7]Jet Propulsion Laboratory (JPL), Pasadena, CA 91109, USA

*Correspondence to*: Michael Buchwitz (buchwitz@uni-bremen.de)

**Abstract.** The COVID-19 pandemic resulted in reduced anthropogenic carbon dioxide ($CO_2$) emissions during 2020 in large parts of the world. We report results from a first attempt to determine whether a regional-scale reduction of anthropogenic $CO_2$

emissions during the COVID-19 pandemic can be detected using space-based observations of atmospheric $CO_2$. For this purpose, we have analysed a small ensemble of satellite retrievals of column-averaged dry-air mole fractions of $CO_2$, i.e. $XCO_2$. We focus on East China because COVID-19 related $CO_2$ emission reductions are expected to be largest there early in the pandemic. We analysed four $XCO_2$ data products from the satellites Orbiting Carbon Observatory-2 (OCO-2) and Greenhouse gases Observing SATellite (GOSAT). We use a data-driven approach that does not rely on *a priori* information

about $CO_2$ sources and sinks and ignores atmospheric transport. Our approach utilises the computation of $XCO_2$ anomalies, $\Delta XCO_2$, from the satellite Level 2 data products using a method called DAM (Daily Anomalies via (latitude band) Medians). DAM removes large-scale, daily $XCO_2$ background variations, yielding $XCO_2$ anomalies that correlate with the location of major $CO_2$ source regions such as East China. We analysed satellite data between January 2015 and May 2020 and compared monthly $XCO_2$ anomalies in 2020 with corresponding monthly $XCO_2$ anomalies of previous years. In order to link the $XCO_2$

anomalies to East China fossil fuel (FF) emissions, we used $XCO_2$ and corresponding FF emissions from NOAA's (National Oceanic and Atmospheric Administration) CarbonTracker version CT2019 from 2015 to 2018. Using this CT2019 data set,



we found that the relationship between target region $\Delta XCO_2$ and the FF emissions of the target region is approximately linear and we quantified slope and offset via a linear fit. We use the empirically obtained linear equation to compute $\Delta XCO_2^{FF}$, an estimate of the target region FF emissions, from the satellite-derived $XCO_2$ anomalies, $\Delta XCO_2$. We focus on October to May periods to minimize contributions from biospheric carbon fluxes and quantified the error of our FF estimation method for this period by applying it to CT2019. We found that the difference of the retrieved FF emissions and the CT2019 FF emissions in terms of the root-mean-square-error (RMSE) is 0.39 $GtCO_2$/year (4%). We applied our method to NASA's (National Aeronautics and Space Administration) OCO-2 $XCO_2$ data product (version 10r) and to three GOSAT products. We focus on estimates of the relative change of East China monthly emissions in 2020 relative to previous months. Our results show considerable month-to-month variability (especially for the GOSAT products) and significant differences across the ensemble of satellite data products analysed. The ensemble mean indicates emission reductions by approximately 8% ± 10% in March 2020 and 10% ± 10% in April 2020 (uncertainties are 1-sigma) and somewhat lower reductions for the other months in 2020. Using only the OCO-2 data product, we obtain smaller reductions of 1-2% (depending on month) with an uncertainty of ±2%. The large uncertainty and the differences of the results obtained for the individual ensemble members indicates that it is challenging to reliably detect and to accurately quantify the emission reduction. There are several reasons for this including the weak signal (the expected regional $XCO_2$ reduction is only on the order of 0.1-0.2 ppm), the sparseness of the satellite data, remaining biases and limitations of our relatively simple data-driven analysis approach. Inferring COVID-19 related information on regional-scale $CO_2$ emissions using current satellite $XCO_2$ retrievals likely requires, if at all possible, a more sophisticated analysis method including detailed transport modelling and considering *a priori* information on anthropogenic and natural $CO_2$ surface fluxes.

## 1 Introduction

Carbon dioxide ($CO_2$) is the most important anthropogenic greenhouse gas significantly contributing to global warming (IPCC, 2013). $CO_2$ has many natural and anthropogenic sources and sinks and our current understanding of them has significant gaps (e.g., Reuter et al., 2017c; Friedlingstein et al., 2019). Efforts are ongoing to improve the fundamental understanding of the global carbon cycle, to improve our ability to project future changes, and to verify the effectiveness of policies such as the Paris Agreement (https://unfccc.int/process-and-meetings/the-paris-agreement/the-paris-agreement, last access: 8-Sept-2020) aiming to reduce greenhouse gas emissions (e.g., Ciais et al., 2014, 2015; Pinty et al., 2017, 2019; Crisp et al., 2018; Matsunaga and Maksyutov, 2018; Janssens-Maenhout et al., 2020).

Retrievals of $XCO_2$ from the satellite sensors SCIAMACHY/ENVISAT (Burrows et al., 1995; Bovensmann et al., 1999; Reuter et al., 2010, 2011), TANSO-FTS/GOSAT (Kuze et al., 2016) and from the Orbiting Carbon Observatory-2 (OCO-2) satellite (Crisp et al., 2004; Eldering et al., 2017; O'Dell et al., 2012, 2018) have been used in recent years to obtain information on natural $CO_2$ sources and sinks (e.g., Basu et al., 2013; Chevallier et al., 2014, 2015; Reuter et al., 2014a, 2017c; Schneising et al., 2014; Houweling et al., 2015; Kaminski et al., 2017; Liu et al., 2017; Eldering et al., 2017; Yin et al., 2018; Palmer et





al., 2019; Miller and Michalak, 2020), on anthropogenic $CO_2$ emissions (e.g., Schneising et al., 2008, 2013; Reuter et al., 2014b, 2019; Nassar et al., 2017; Schwandner et al., 2017; Matsunaga and Maksyutov, 2018; Miller et al., 2019; Labzovskii et al., 2019; Wu et al., 2020; Zheng et al., 2020a; Ye et al., 2020) and for other applications such as climate model assessments (e.g., Lauer et al., 2017; Gier et al., 2020) or data assimilation (e.g., Massart et al., 2016).

Here we use an ensemble of satellite retrievals of $XCO_2$ to determine whether COVID-19 related regional-scale $CO_2$ emission
reductions can be detected and quantified using the current space-based observing system. This is important in order to establish the capabilities of current satellites, which have been optimized to obtain information on natural carbon sources and sinks, but not to obtain information on anthropogenic emissions. Nevertheless, data from existing satellites have already been used to assess anthropogenic emissions (see publications cited above). These assessments and the assessment presented in this publication are relevant for future satellites focussing on anthropogenic emissions such as the planned European Copernicus
Anthropogenic $CO_2$ Monitoring (CO2M) mission (e.g., ESA, 2019; Kuhlmann et al., 2019;  Janssens-Maenhout et al., 2020), which is based on the CarbonSat concept (Bovensmann et al., 2010; Velazco et al., 2011; Buchwitz et al., 2013; Pillai et al., 2016; Broquet et al., 2018).

We focus on China because regional-scale COVID-19 related $CO_2$ emission reductions are expected to be largest there early in the pandemic (Le Quéré et al., 2020; Liu et al., 2020). Satellite data have been used to estimate China's $CO_2$ emissions
during the COVID-19 pandemic (Zheng et al., 2020b), but that study inferred $CO_2$ reductions from retrievals of nitrogen dioxide ($NO_2$) not using $XCO_2$. Estimates of emission reductions have also been derived from bottom-up statistical assessments of fossil fuel use and other economic indicators. According to Le Quéré et al., 2020, China's $CO_2$ emissions decreased by 242 $MtCO_2$ (uncertainty range 108 – 394 $MtCO_2$) during January – April 2020. As China's annual $CO_2$ emissions are approximately 10 $GtCO_2$/year (Friedlingstein et al., 2019), i.e., approximately 3.3 $GtCO_2$ in a 4-month period assuming constant emissions,
the average relative (COVID-19 related) change during January – April 2020 is therefore approximately 7% ± 4% (0.242/3.3 ± 0.14/3.3). This agrees reasonably well with the estimate reported in Liu et al., (2020), which is 10.3% for China during the first quarter of 2020 compared to the same period in 2019. Liu et al., (2020) also indicate some challenges in terms of interpreting $CO_2$ emission reductions as being caused by COVID-19, e.g., the fact that the first months of 2020 were exceptionally warm across much of the northern hemisphere. $CO_2$ emissions associated with home heating may have therefore
been somewhat lower than for the same period in 2019, even without the disruption in economic activities and energy production caused by COVID-19 and related lockdowns.

Sussmann and Rettinger, 2020, studied grond-based remote sensing $XCO_2$ retrievals of the Total Carbon Column Observing Network (TCCON) to find out whether related atmospheric concentration changes may be detected by the TCCON and brought into agreement with bottom-up emission-reduction estimates. To the best of our knowledge, our study is the first
attempt to determine quantitatively whether COVID-19 related regional-scale $CO_2$ emission reductions can be detected using existing space-based observations of $XCO_2$, although some qualitative results related to this application are also provided in the internet (e.g., ESA-NASA-JAXA, 2020). Regional-scale reductions of tropospheric $NO_2$ columns have been reported for



China (e.g., Zhang et al., 2020; Bauwens et al., 2020), but for $CO_2$ such an assessment is more challenging because of small $XCO_2$ changes on top of a large background. For example, over extended anthropogenic source areas such as East China, the
$XCO_2$ enhancement due to anthropogenic emissions is typically only approximately 1 - 2 ppm (0.25% - 0.5% of 400 ppm) or even less (see, e.g., Schneising et al., 2008, 2013; Hakkarainen et al., 2016, 2019; and this study). A 10% emission reduction would therefore only change the regional $XCO_2$ enhancement by 0.1 to 0.2 ppm. This is below the single measurement precision of current satellite $XCO_2$ data products, which is about 1.8 ppm (1-sigma) (e.g., Dils et al., 2014; Kulawik et al., 2016; Buchwitz et al., 2015, 2017a; Reuter et al., 2020) for GOSAT and around 1 ppm for OCO-2 (Wunch et al., 2017;
Reuter et al., 2019). This implies that the satellite data need to be averaged to reduce random error (noise) contributions. Therefore, we focus on $XCO_2$ monthly averages. The accuracy of the East China satellite $XCO_2$ retrievals averaged over monthly timescales is difficult to assess because of limited reference data. The validation of the satellite data products is primarily based on comparisons with ground-based $XCO_2$ retrievals from the TCCON, a relatively sparse network with an uncertainty of about 0.4 ppm (Wunch et al., 2010). The estimated precision and accuracy of the satellite $XCO_2$ retrievals as
obtained from comparisons with TCCON $XCO_2$ retrievals is typically on the order of 0.7 ppm (e.g., Buchwitz et al., 2017a; Reuter et al., 2020) but this estimate assumes error-free TCCON retrievals, i.e., it neglects the non-negligible uncertainty of TCCON.

This manuscript is structured as follows: In Sect. 2 we present the satellite and model data used for this study and in Sect. 3 we present the analysis method. The main section is Sect. 4 where we present and discuss the results. A summary and
conclusions are given in Sect. 5.

## 2 Data

In this section, we present a short overview about the input data used for this study.

### 2.1 Satellite $XCO_2$ estimates

This study uses four satellite $XCO_2$ Level 2 (L2) data products. An overview about these data sets is provided in Tab. 1 including references and access information. The first product listed in Tab. 1 is the latest bias-corrected OCO-2 $XCO_2$ product delivered to the Goddard Earth Science Data and Information Services Center (GES DISC) by the OCO-2 team (ACOS v10r Lite). The other three satellite $XCO_2$ datasets are different versions of the GOSAT $XCO_2$ product derived using retrieval algorithms developed by groups at the University of Leicester, U.K. (UoL-FP v7.3), the Netherlands Institute for Space
Research (RemoTeC v2.3.8), and the University of Bremen, Germany (FOCAL v1.0).

The $XCO_2$ estimates derived from OCO-2 and GOSAT observations are complementary because these two spacecraft use different sampling strategies. OCO-2 has been operating since September 2014. Its spectrometers collect about 85000 cloud-free $XCO_2$ soundings each day along a narrow (< 10 km) ground track as it orbits the Earth 14.5 times each day from its sun synchronous 1:36 PM orbit. The OCO-2 soundings provide continuous measurements with relatively high spatial resolution



(2.25 km) along each track, but the individual ground tracks are separated by almost 25° longitude in any given day. This spacing is reduced to approximately 1.5° longitude after a 16-day ground track repeat cycle. GOSAT has been returning 300 to 1000 cloud-free $XCO_2$ soundings each day since April 2009. Its TANSO-FTS spectrometer collects soundings with ~10.6 km diameter surface footprints, separated by approximately 250 km along and across its ground track at it orbits from north to south across the sunlit hemisphere.

**2.2 Model $CO_2$ data**

We use data from NOAA's (National Oceanic and Atmospheric Administration) $CO_2$ assimilation system, CarbonTracker (CT2019) (Jacobson et al., 2020; Peters et al., 2007) to define the relationship between $XCO_2$ anomalies and fossil fuel emissions. CarbonTracker is a global atmospheric inverse model that assimilates atmospheric $CO_2$ measurements as well as estimates of emissions from fossil fuels and fires and other sources into an atmospheric transport model to estimate emissions

and uptake of $CO_2$ by the land biosphere and oceans. An overview about CT2019 set is provided in Tab. 2 including references and access information. A description of CT2019 can also found on the CT2019 website (https://www.esrl.noaa.gov/gmd/ccgg/carbontracker/index.php, last access: 24-September 2020): "CarbonTracker produces model predictions of atmospheric $CO_2$ mole fractions, to be compared with the observed atmospheric $CO_2$ mole fractions. The difference between them is attributed to differences in the sources and sinks used to make the prediction (the so-called 'first-

guess') and the sources and sinks affecting the true atmospheric $CO_2$. Using numerical techniques, these differences are used to solve for a set of sources and sinks that most closely matches the observed $CO_2$ in the atmosphere. CarbonTracker has a representation of atmospheric transport based on weather forecasts, and modules representing air-sea exchange of $CO_2$, photosynthesis and respiration by the terrestrial biosphere, and release of $CO_2$ to the atmosphere by fires and combustion of fossil fuels."


**3. Methods**

To analyse the satellite data with respect to regionally elevated $XCO_2$ due to anthropogenic $CO_2$ emissions, we use a method referred to as DAM (Daily Anomalies via (latitude band) Medians). The DAM method is essentially identical with the

method described in Hakkarainen et al., 2016 and 2019. They applied their method to the OCO-2 Level 2 $XCO_2$ data product to filter out trends and seasonal variations in order to isolate $CO_2$ source/sink signals. Hakkarainen et al., 2019, explain their method as follows: "In order to obtain the background, we calculate the daily medians for each 10-degree latitude band and linearly interpolate the resulting values to each OCO-2 data point. The median was chosen because it better represents the typical value in each latitude band, and it is not skewed towards extreme values". Our approach is very similar, but instead

of interpolation, we compute the median around each latitude ("running median") using a latitude band width of ±15 deg. We use a larger width compared to Hakkarainen et al., 2019, because we also apply our method to GOSAT data, which are much sparser than OCO-2 data. Our investigations showed that the width of the latitude band is not critical but needs to be



wide enough to contain a statistically significant sample, but narrow enough to resolve large latitudinal gradients in $CO_2$. We subtract the corresponding median from each single $XCO_2$ observation as contained in the original Level 2 $XCO_2$ data

product files to obtain a data set of $XCO_2$ anomalies, $\Delta XCO_2$. For illustration, we gridded these anomalies to obtain global maps. Figure 1 shows such a DAM $XCO_2$ anomaly map at $1^{\circ}$x$1^{\circ}$ resolution covering the time period 2015 – 2019. The resulting spatially resolved $XCO_2$ anomalies are very similar as the one shown in Hakkarainen et al., 2019 (see their Fig. 3, top panel). The good agreement confirms the finding reported above that the generation of these anomaly maps does not critically depend on how exactly the median is computed and used to subtract "the background".

A zoom into Fig. 1 is presented in Fig. 2, which shows more details for China and surrounding areas. As can be seen from Fig. 2, the DAM $\Delta XCO_2$ has a positive anomaly especially in the region between Beijing, Wuhan and Hong Kong with highest values in the area between Beijing and Shanghai. This positive anomaly indicates that this region is an important $CO_2$ source region. Of course, there is no one-to-one correspondence (especially not for every grid cell) between these $XCO_2$ anomalies and local $CO_2$ emissions (or uptake) because the emitted $CO_2$ is transported and mixed in the atmosphere.

Furthermore, the satellite data are typically sparse due to strict quality filtering to avoid potential $XCO_2$ biases, for example, due to the presence of clouds. Cloud contaminated ground scenes are identified to the extent possible via the corresponding retrieval algorithm and flagged to be "bad" (see references listed in Tab. 1) and are therefore not used for this analysis. The sparseness of the satellite data set is obvious from Fig. 3, which shows DAM $XCO_2$ anomaly maps for the month of February during the six years from 2015 to 2020.

The geographical coordinates of the East China target region investigated here are listed in Tab. 3. The fossil fuel (FF) $CO_2$ emissions of this target region are approximately 8 GtCO$_2$/year, i.e., the selected target region covers approximately 80% of the FF emissions of entire China, which are approximately 10 GtCO$_2$/year (Le Quéré et al., 2018; Friedlingstein et al., 2019).

For our East China analysis, we compute regional averages of DAM $XCO_2$ anomalies by averaging the $\Delta XCO_2$ values for all satellite ground scenes (footprints) as located in the East China target region for each day in the period January 2015 to May

2020. These East China daily time series are then further averaged to obtain monthly East China $\Delta XCO_2$ as presented in the following section.

## 4. Results and discussion


### 4.1 Application of the DAM method to model data

To determine whether satellite $XCO_2$ retrievals can provide information on relative changes of anthropogenic $CO_2$ emissions for the East China target region, we must establish the relationship between the DAM $XCO_2$ anomalies (see Sect. 3) and the target region fossil fuel (FF) emissions. For this purpose, we use the CarbonTracker (CT2019) data set described in Sect. 2.2.



Figure 4 shows CT2019 $XCO_2$ maps (left) and corresponding surface $CO_2$ flux maps (right) for selected days in the January to May 2018 period. The model data are sampled at local noon, which is close to the overpass time of the satellite data sets used here. As can be seen from Fig. 4, $XCO_2$ is clearly elevated over the East China target region (red rectangle) relative to its surrounding region on 15-January-2018 (panel (a)) and on 15-March-2018 (panel (c)). On 15-May-2018 (panel (e)) the target region and parts of the surrounding region contain large areas of lower than average $XCO_2$, a pattern which primarily results

from carbon uptake by vegetation during the growing season, which starts around May each year. The $CO_2$ fluxes, which are shown in the right-hand side panels of Fig. 4, show similar spatial pattern as the $XCO_2$ maps but as already explained, there is not a one-to-one correspondence between atmospheric $XCO_2$ and surface emissions due to atmospheric transport. The $CO_2$ fluxes are the sum of several contributing fluxes including FF emissions, biospheric fluxes and other fluxes (e.g., due to fires and the oceans).

Figure 5 shows time series obtained by applying the DAM method to CT2019 $XCO_2$ for the East China target region. The top panel shows the CT2019 FF emissions as thick red line. For each day, the DAM method was applied to CT2019 $XCO_2$ to compute daily (and then monthly) $XCO_2$ anomalies, $\Delta XCO_2$. The monthly CT2019 $\Delta XCO_2$ values were linearly fitted to the monthly CT2019 FF emissions to obtain a quantity referred to as $\Delta XCO_2^{FF}$, which closely resembles the FF emissions as shown in the top panel of Fig. 5. The linear fit yields $\Delta XCO_2^{FF} = 0.914 \times \Delta XCO_2 + 7.106$, with $\Delta XCO_2$ in parts per million

(ppm) and $\Delta XCO_2^{FF}$ in GtCO$_2$/year. Using this linear transformation, $\Delta XCO_2$ can be transformed into $\Delta XCO_2^{FF}$ and daily and monthly $\Delta XCO_2^{FF}$ values are shown in Fig. 5 as thin grey line and blue dots, respectively.

Initially, we assumed that $\Delta XCO_2$ is directly proportional to the target region fossil fuel emissions, i.e., we assumed that FF is (approximately) equal to a constant multiplied by $\Delta XCO_2$ (no offset added). If we were only interested in relative changes in emissions, we would not need the exact values of the scaling factor. We found that $\Delta XCO_2$ is around 2 ppm for January but

decreases in subsequent months, nearly approaching zero in May (not shown here but see Fig. 5, top panel, showing a time series of $\Delta XCO_2^{FF}$ (blue dots)). As anthropogenic emissions are not expected to change that much within a few months we concluded that the simple proportionality assumption does not hold. We then used the CT2019 data set to test our method. We applied our method to CT2019 $XCO_2$ and compared the retrieved FF values with the true FF values as used by CT2019. We found large differences which could be significantly reduced by adding an offset. To obtain numerical values for the offset

(see parameter B in Fig. 5, top panel) and for the scaling factor (see parameter A in Fig. 5, top panel) we used a linear fit as explained above, i.e., these two parameters are obtained empirically.

We use target region monthly $\Delta XCO_2^{FF}$ as a satellite-based estimate of target region monthly FF emissions. The middle and the bottom panels of Fig. 5 show the absolute and the relative monthly differences between $\Delta XCO_2^{FF}$ and the FF emissions, respectively. Also listed are several quantities used to characterise the level of agreement/disagreement: D is the mean

difference, S is the standard deviation of the difference, RMSE is the root-mean-square-error and R is the linear correlation coefficient of the monthly $\Delta XCO_2^{FF}$ and FF values. As can be seen, the RMSE is 0.45 GtCO$_2$/year (6%) for the entire time series 2015 – 2018. The RMSE is reduced to 0.39 GtCO$_2$/year (4%) if the analysis is restricted to time periods covered by the





months October to May, which is the relevant period for this study as it covers pre-COVID-19 (October – December 2019) and COVID-19 (January – May 2020) periods. Excluding June to September data reduces the RMSE because (during this

summer period) disturbances from biospheric fluxes (i.e., photosynthesis related fluxes during the vegetation growing season) are largest. As shown in Fig. 5, $\Delta XCO_2^{FF}$ is to a good approximation proportional to FF emissions and for this study it is assumed that relative changes of the monthly $\Delta XCO_2^{FF}$ values can be used as a sufficiently accurate proxy for relative changes of the monthly FF emissions.

The method described in this section has been applied to convert satellite-derived target region $XCO_2$ anomalies, $\Delta XCO_2$, into

$\Delta XCO_2^{FF}$. Monthly $\Delta XCO_2^{FF}$ values have been computed for each satellite data product and used to find out if FF reductions during the COVID-19 pandemic can be detected in these time series, as will be explained in the following sub-sections.

## 4.2 Application of the DAM method to satellite XCO₂ retrievals

The DAM method described in the previous section has been applied to a small ensemble of $XCO_2$ retrievals from OCO-2 and GOSAT (see details listed in Tab. 1) and the results are presented below. As noted above, a key difference between the OCO-2 and the GOSAT data products is the different sampling of the target region, with GOSAT having much sparser coverage compared to OCO-2. This is illustrated in Fig. 6, which shows February to March 2020 averages of the OCO-2 $XCO_2$ data product (Fig. 6a) and the three GOSAT data products (Fig. 6b – 6d) at 1°x1° resolution. The OCO-2 product shown in Fig. 6a

is NASA's OCO-2 operational "Atmospheric $CO_2$ Observations from Space" (ACOS) algorithm version 10r bias corrected $XCO_2$ product (the so called Lite product), which is referred to in this publication via the product identifier (ID) CO2_OC2_ACOS. The three GOSAT $XCO_2$ products are (see details and references as given in Tab. 1): Fig. 6b: University of Leicester's GOSAT product (ID CO2_GOS_OCFP); Fig. 6c: SRON Netherlands Institute for Space Research GOSAT product (CO2_GOS_SRFP); Fig. 6d: University of Bremen's GOSAT product (CO2_GOS_FOCA) as retrieved with the "Fast

atmOspheric traCe gAs retrievaL" (FOCAL) retrieval algorithm initially developed for OCO-2 (Reuter et al., 2017a, 2017b). As can be seen from Fig. 6, the spatial sampling of the target region is different for each product as only quality-filtered (i.e., "good") data are shown and the quality filtering is algorithm specific (see references listed in Tab. 1).

## 4.2.1 Application to NASA's OCO-2 (version 10r) XCO₂

Figure 7 shows the results obtained by applying the DAM method to product CO2_OC2_ACOS for the East China target region for the period January 2015 to May 2020. The top panel shows the daily DAM $XCO_2$ anomalies as thin grey line and the corresponding monthly averages as red dots. The amplitude (approximately ±1 ppm) and time dependence (e.g., there is a minimum in the middle of each year) is as expected as can be concluded from a comparison with the corresponding CT2019 results shown in Fig. 5 (but note that Fig. 7 shows $\Delta XCO_2$ whereas Fig. 5 shows $\Delta XCO_2^{FF}$, which differs somewhat in



amplitude and offset due to the applied linear transformation as explained in Sect. 4.1). Two criteria need to be fulfilled for the monthly data shown in this figure: (i) The minimum number of days per month is 5 and (ii) the minimum number of observations per day is 30, i.e., at least 5 days per month having (for each day) at least 30 observations per day in the target region need to be available for a month to be accepted (results for other combinations of these two parameters are presented below).

The middle panel of Fig. 7 shows monthly $\Delta XCO_2^{FF}$ for October - May for five different periods: October 2019 – May 2020 (red), October 2018 – May 2019 (blue) and three corresponding periods in earlier years (see annotation as listed in the figure panel). As can be seen from this panel, the red curve (October 2019 – May 2020) shows, apart from March 2020, somewhat lower values in the period February 2020 – May 2020 (i.e., during the "COVID-19 period") compared to January 2020 and earlier months ("pre-COVID-19 period"). However, the time dependence shows significant month-to-month variability.

Furthermore, a similar time dependency is also present for October 2015 – May 2016 (pink). In comparison to the other periods from earlier years, the (red) October 2019 – May 2020 values are mostly close to the maximum (or even exceed) the values of the other curves corresponding to earlier October – May time periods.

The bottom panel of Figure 7 shows the corresponding relative differences, as we are mostly interested in relative (percentage) changes of the target region FF emissions. The blue dots (and connecting lines) show the relative difference between the red

dots from in the middle panel (this time series ends in May 2020) and the blue dots shown in the middle panel (this time series ends in May 2019). If for simplicity we refer to time series ending in May as "2020" (red dots in middle panel) and "2019" (blue dots in middle panel), then the blue dots displayed in the bottom panel show "(2020-2019)/2019", i.e., they are a proxy for the relative change of the target region FF emissions of the corresponding months in 2020 relative to 2019 (for January to May; for October to December the difference corresponds to 2019 relative to 2018). The green dots show the corresponding

relative differences for 2020 relative to 2018, the orange dots correspond to the relative differences for 2020 relative to 2017 and the pink dots show the relative differences for 2020 relative to 2016. As can be seen from the bottom panel of Fig. 7, the relative difference between the time series ending in May 2020 and in May 2019 (blue dots in the bottom panel) also shows significant variability from month to month. Compared to all previous differences (2020-2018, 2020-2017, etc.) one sees that these differences are mostly positive and typically somewhere in the range between 0% and +10%. Figure 7 indicates that

during October 2019 to May 2020 target region FF emissions are on average a few percent higher compared to previous years.

The data shown in the bottom panel of Fig. 7 are also shown in Fig. 8 (see thin lines using different colours) but together with derived monthly median (and mean) values. In addition, vertical bars are shown to indicate the scatter of the monthly values. As we are primarily interested in changes during the January to May 2020 period compared to preceding months (October – December 2019) and compared to previous years, the median has been computed for each month and the October – December

2019 (i.e., pre-COVID-19 OND months) mean value has been subtracted to highlight the difference of 2020 relative to 2019.

The corresponding monthly medians are shown as thick royal blue symbols (and connecting lines) and the scatter - as computed from the standard deviation of the monthly values – is shown as royal blue vertical bars. The light blue symbols and lines show the corresponding values when using the mean instead of the median. In grey, the "original" median values are shown, i.e., in grey, the median values are shown before the offset has been subtracted (i.e., the "absolute" values are shown in grey and the corresponding "OND anomaly" is shown in royal blue). The royal blue curve indicates that $\Delta XCO_2^{FF}$, our proxy for the target region FF emissions, is 2-3% lower in February to April 2020 compared to October to December 2019 and earlier years. Due to the large scatter and significant month-to-month variability and possibly also for other reasons (e.g., the unusual meteorological conditions in the first few months of 2020, see, for example, Liu et al., 2020) it is unclear to what extent this reduction is related to COVID-19 countermeasures.

Figures 7 and 8 have been generated with the requirement that, for each day, at least 30 observations need to be available in the target region and that for each month, at least 5 days fulfilling this 30 observations/day requirement are available. Figure 9 is similar to Fig. 8 except that results for additional combinations of minimum number of observations per day and minimum number of days per month have been added. Here, the results depend somewhat on which combination of these parameters is used, but the overall end result as shown via the royal blue symbols and lines is fairly stable as it depends only marginally on which set of combinations is used as can be seen when comparing Fig. 9 with Fig. 8.

### 4.2.2 Application to GOSAT XCO₂ data products

The same analysis method as applied to NASA's OCO-2 data product (see Sect. 4.2.1) has also been applied to the three GOSAT XCO₂ data products listed in Tab. 1. The results are shown in Fig. 10 for product CO2_GOS_OCFP, in Fig. 11 for product CO2_GOS_SRFP, and in Fig. 12 for product CO2_GOS_FOCA.

The month-to-month variations are larger for these products compared to product CO2_OC2_ACOS (e.g., Figs. 9; note the different scales of the y-axes). This is very likely because GOSAT products are sparse compared to the OCO-2 product (see Fig. 6) but also because the single observation random error (precision) is larger for GOSAT.



Analysis of product CO2_GOS_OCFP (Fig. 10) suggests that on average, emissions are reduced in 2020 (approximately -

12% ± 12%) but strong conclusions cannot be drawn because of large uncertainty (approximately 12%, 1-sigma). Product

CO2_GOS_SRFP (Fig. 11) shows no clear time dependence due to large month-to-month variability and the same seems to

be true for product CO2_GOS_FOCA (Fig. 12).

### 4.2.3 Ensemble mean and uncertainty

An overview about the results obtained from all four satellite data products is shown in Fig. 13. The results obtained from

the individual products (as shown in royal blue in Figs. 9 - 12) are shown here using reddish colours (the corresponding

numerical values are listed in Tab. 4). Also shown in Fig. 13 is the mean of the ensemble members and its estimated

uncertainty (in dark blue); the corresponding numerical values are listed in the bottom row of Tab. 4. In Fig. 13, the

ensemble mean shows on average slightly lower values during March and April 2020 compared to the other months,

suggesting an emission reduction by several percent (-8% in March and -10% in April). However, the 1-sigma uncertainties

shown as dark blue vertical lines are large (±10%) and typically overlap with the zero, i.e., no change, line. Furthermore, as

already discussed, there are considerable month-to-month variations and large differences between the results obtained from

the individual satellite data products. It is therefore concluded that the expected reduction cannot be accurately quantified.

### 5 Summary and conclusions


We have analysed a small ensemble of retrieved satellite $XCO_2$ data products to investigate whether a regional-scale reduction

of atmospheric $CO_2$ during the COVID-19 pandemic can be detected over East China. Specifically, we analysed four $XCO_2$

data products from the satellites OCO-2 and GOSAT. For this purpose, we used a data-driven approach, which involves the

computation of $XCO_2$ anomalies, $\Delta XCO_2$, using a method called DAM (Daily Anomalies via (latitude band) Medians). This

method, which is essentially identical with the method developed at Finnish Meteorological Institute (FMI, Hakkarainen et al.,

2019), helps to isolate local or regional $XCO_2$ enhancements originating from anthropogenic $CO_2$ emissions from large-scale

daily $XCO_2$ background variations. We analysed satellite data between January 2015 to May 2020 and compared year 2020

monthly $XCO_2$ anomalies with the corresponding monthly $XCO_2$ anomalies from previous months.

In order to link the satellite-derived $XCO_2$ anomalies to East China fossil fuel (FF) emissions, we used output from NOAA's

$CO_2$ assimilation system CarbonTracker (CT2019). Using CT2019, we show that $\Delta XCO_2$ can be linearly transformed to "FF





related $XCO_2$ enhancements", denoted $\Delta XCO_2^{FF}$, and via a linear fit we established a linear empirical equation to relate these two quantities. We use this empirical equation to compute $\Delta XCO_2^{FF}$, an estimate of the target region FF emissions, from the satellite-derived $XCO_2$ anomalies, $\Delta XCO_2$. We focus on October to May periods and found using CT2019 that the root-mean-square-error (RMSE) of our FF estimation method is approximately 0.39 GtCO2/year (4%). These months are less affected by

disturbances from natural terrestrial vegetation carbon fluxes (using all months the RMSE is larger, namely 0.45 GtCO2/year or 6%) and appropriate as they cover relevant pre-COVID-19 (e.g., October – December 2019) and COVID-19 (January – May 2020) periods. However, satellite retrievals during these months are more challenging because of the low sun angles and cloudiness.

We applied our method to each of the four satellite $XCO_2$ data products and computed monthly $\Delta XCO_2^{FF}$ for East China

("retrieved East China monthly FF emissions"). Our results show considerable month-to-month variability (especially for the GOSAT products) and significant differences across the ensemble of satellite data products analysed. The ensemble mean suggests emission reductions by approximately 8% ± 10% in March 2020 and 10% ± 10% in April 2020 (uncertainties are 1-sigma) and somewhat lower reductions for the other months in 2020. These values are dominated by the GOSAT ensemble members. Analysis of the OCO-2 product yields smaller values showing a reduction of 1-2% with an uncertainty of ±2%.

The large uncertainty, which is on the order of the derived reduction (i.e., 100%, 1-sigma), and the large spread of the results obtained for the individual ensemble members, indicates that it is challenging to reliably detect and to accurately quantify the emission reduction using the current generation of space based methods and the simple DAM analysis strategy adopted here. These findings are not unexpected. Fossil fuel emissions related regional $XCO_2$ enhancements are typically only 1 to 2 ppm and even a 10% emission reduction would therefore only correspond to a reduction of the fossil fuel related regional $XCO_2$

enhancement by 0.1 to 0.2 ppm. $XCO_2$ variations as small as 0.2 ppm are below the estimated uncertainty of the single footprint satellite $XCO_2$ retrievals. This single observations uncertainty, which is around 0.7 ppm (e.g., Buchwitz et al., 2017a; Reuter et al., 2020), has been obtained by comparisons with ground-based Total Carbon Column Observing Network (TCCON) $XCO_2$ retrievals, which have an uncertainty of 0.4 ppm (1-sigma, Wunch et al., 2010). To reduce random errors, we use monthly averaged data. Averaging results in reducing the random error but investigations have shown that random errors do not simply

scale with the inverse of the square root of number of observations added (Kulawik et al., 2016).

We conclude that inferring COVID-19 related information on regional-scale $CO_2$ emissions using current (quite sparse) satellite $XCO_2$ retrievals requires, if at all possible, a more sophisticated analysis method including the use of detailed *a priori* information and atmospheric transport modelling.

The extent to which COVID-19 related emission reductions can be resolved on smaller scales - such as power plants or cities

(e.g., Nassar et al., 2017; Reuter et al., 2019; Zheng et al., 2020a; Wu et al., 2020) has not yet been investigated in detail (to the best of our knowledge). For this purpose, $XCO_2$ retrievals from NASA's OCO-3 mission are also very promising, especially because of its Snapshot Area Map (SAM) mode, which permits the mapping of $XCO_2$ over ~80 km by 80 km areas around





localized anthropogenic $CO_2$ emission sources (see https://ocov3.jpl.nasa.gov/ (last access: 28-Aug-2020)). Even more complete coverage is planned for the Copernicus CO2M mission in the future (e.g., Janssens-Maenhout et al., 2020).




## Acknowledgements

This study has been funded in parts by the European Space Agency (ESA) via projects ICOVAC (Impacts of COVID-19 lockdown measures on Air quality and Climate) and GHG-CCI+ (http://cci.esa.int/ghg, last access: 13-August-2020) and the University and the State of Bremen. We acknowledge financial support for the generation of several data sets used as input

for this study: (i) European Commission via Copernicus Climate Change Service (C3S, https://climate.copernicus.eu/, last access: 22-July-2020) project C3S_312b_Lot2, (ii) the Japanese space agency JAXA (contract 19RT000692) and (iii) EUMETSAT (contract EUM/CO/19/4600002372/RL). H.Boe., Univ. Leicester, was funded as part of NERC's support of the National Centre for Earth Observation (NE/R016518/1).

We also acknowledge access to OCO-2 XCO$_2$ data product "OCO2_L2_Lite_FP 10r" obtained from NASA's Earthdata GES

DISC website (https://disc.gsfc.nasa.gov/datasets?keywords=OCO-2%20v10r&page=1 (last access: 15-Aug-2020)).

We thank JAXA and the National Institute for Environmental Studies (NIES), Japan, for access to GOSAT Level 1 (L1) data and ESA for making the GOSAT L1 products available via the ESA Third Party Mission (TPM) archive.

We acknowledge CarbonTracker CT2019 results provided by NOAA ESRL, Boulder, Colorado, USA, from the website at http://carbontracker.noaa.gov (last access: 22-July-2020). We also acknowledge feedback from Andy Jacobson on an early

draft of this manuscript.

Some of the work reported here was conducted by the Jet Propulsion Laboratory, California Institute of Technology under contract to NASA. Government sponsorship is acknowledged.

## Author contributions

M.B. designed the study, performed the analysis and led the writing of this paper in close cooperation with M.R., S.N., B.F.A., H.B., J.P.B., O.S. K.B. and M.H. Input data and corresponding advice has been provided by M.R., S.N., A.D.N., H.Boe., L.W, J. L., I.A., C.W.O'D. and D.C. All authors contributed to significantly improve the manuscript.

**Data availability.** The key results of this study are listed in this manuscript in numerical form (Tab. 4). Access information
for the satellite data used as input for this study is provided in Tab. 1. The CT2019 data are available from NOAA (see access information given in Tab. 2).

## Competing financial interests

The authors declare no competing financial interests.



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





**Tables:**

**Table 1. Overview of the satellite XCO₂ Level 2 (L2) input data products. (#) These products are available via the Copernicus Climate Data Store (CDS, https://cds.climate.copernicus.eu/cdsapp#!/dataset/satellite-carbon-dioxide?tab=overview (last access: 23-September-2020)) until end of 2019. Year 2020 data will be made available via the CDS in 2021 but are also available from the authors on request.**

| Satellite | Algorithm | Product version | Product ID | References | Data provider and data access information |
|-----------|-----------|-----------------|------------|------------|-------------------------------------------|
| OCO-2 | ACOS | v10r | CO2_OC2_ACOS | O'Dell et al., 2018; Kiel et al., 2019; Osterman et al., 2020 | Product "OCO2_L2_Lite_FP 10r" obtained from NASA's Earthdata GES DISC website: https://disc.gsfc.nasa.gov/datasets?keywords=OCO-2%20v10r&page=1 (last access: 15-Aug-2020) |
| GOSAT | UoL-FP | v7.3 | CO2_GOS_OCFP | Cogan et al., 2012; Boesch et al., 2019 | Generated by authors (#) |
| GOSAT | RemoTeC | v2.3.8 | CO2_GOS_SRFP | Butz et al., 2011; Wu et al., 2019 | Generated by authors (#) |
| GOSAT | FOCAL | v1.0 | CO2_GOS_FOCA | Noël et al., 2020 | Generated by authors and available on request |



**Table 2. Overview of the CarbonTracker CT2019 data set. For this study we used data from the period January 2015 to December 2018.**

| Model / Version | Details | Reference | Access |
|---|---|---|---|
| CarbonTracker CT2019 | Atmospheric $CO_2$ molefraction profiles (spatio-temporal sampling: 3°x2°, 3-hourly) and $CO_2$ fluxes (spatio-temporal sampling: 1°x1°, 3-hourly) | Jacobson et al., 2020 <br><br> DOI: <br><br> http://dx.doi.org/10.25925/39m3-6069 <br><br> (last access: 22-July-2020) | CarbonTracker CT2019, <br><br> http://carbontracker.noaa.gov <br><br> (last access: 22-July-2020) |


**Table 3. Corner coordinates of the East China target region as analysed in this study.**

| Region ID | Latitude range [deg North] | Lontitude range [deg East] |
|---|---|---|
| **East China** | 28 – 44 | 102 – 126 |






**Table 4. Numerical values of the ensemble-based ΔXCO$_2$$^{FF}$ results as shown in Fig. 13. Listed are the median values and corresponding 1-sigma uncertainties (in brackets). The dimensionless values listed here represent the relative ΔXCO$_2$$^{FF}$ change for**

**January-May 2020 relative to October-December 2019 and previous years. The listed data refer to the difference relative to October to December 2019, i.e., the corresponding offset (October to December 2019 mean) has been subtracted.**

| Month<br>Product ID | October<br>2019 | November<br>2019 | December<br>2019 | January<br>2020 | February<br>2020 | March<br>2020 | April<br>2020 | May<br>2020 |
|---|---|---|---|---|---|---|---|---|
| **CO2_OC2_ACOS** | 0.000<br>(0.023) | 0.005<br>(0.032) | -0.005<br>(0.018) | 0.015<br>(0.029) | -0.008<br>(0.020) | -0.007<br>(0.016) | -0.023<br>(0.017) | -0.016<br>(0.028) |
| **CO2_GOS_OCFP** | -0.084<br>(0.077) | 0.025<br>(0.074) | 0.058<br>(0.053) | -0.133<br>(0.035) | -0.056<br>(0.051) | -0.151<br>(0.087) | -0.242<br>(0.031) | -0.110<br>(0.160) |
| **CO2_GOS_SRFP** | -0.067<br>(0.039) | 0.130<br>(0.045) | -0.063<br>(0.049) | 0.043<br>(0.091) | -0.064<br>(0.061) | 0.024<br>(0.093) | -0.106<br>(0.046) | 0.010<br>(0.082) |
| **CO2_GOS_FOCA** | -0.048<br>(0.049) | 0.052<br>(0.062) | -0.004<br>(0.043) | -0.041<br>(0.042) | -0.016<br>(0.104) | -0.189<br>(0.057) | -0.040<br>(0.070) | -0.110<br>(0.063) |
| **Ensemble** | -0.050<br>(0.036) | 0.053<br>(0.055) | -0.004<br>(0.049) | -0.029<br>(0.078) | -0.036<br>(0.028) | -0.081<br>(0.105) | -0.103<br>(0.099) | -0.057<br>(0.063) |






**Figures:**


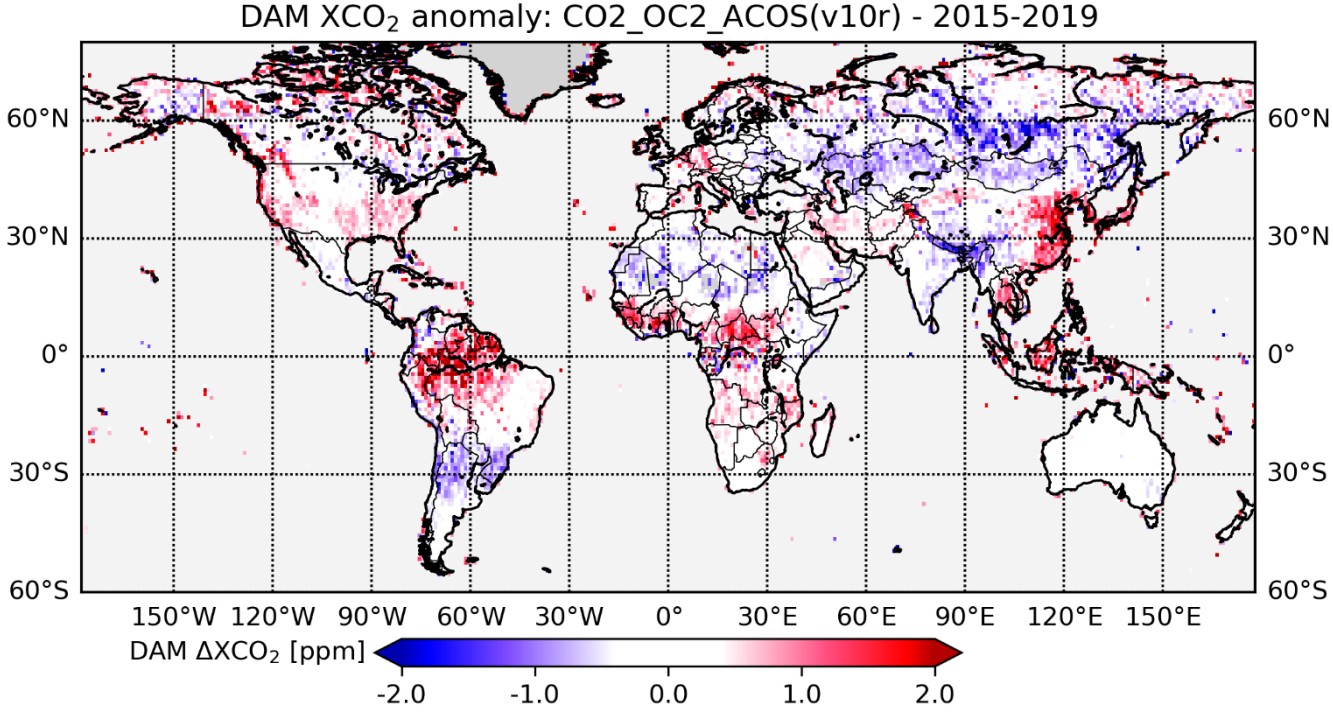

**Figure 1: DAM XCO₂ anomaly map at 1º x 1º resolution generated from OCO-2 Level 2 XCO₂ (v10r, land) for 2015 to 2019.**




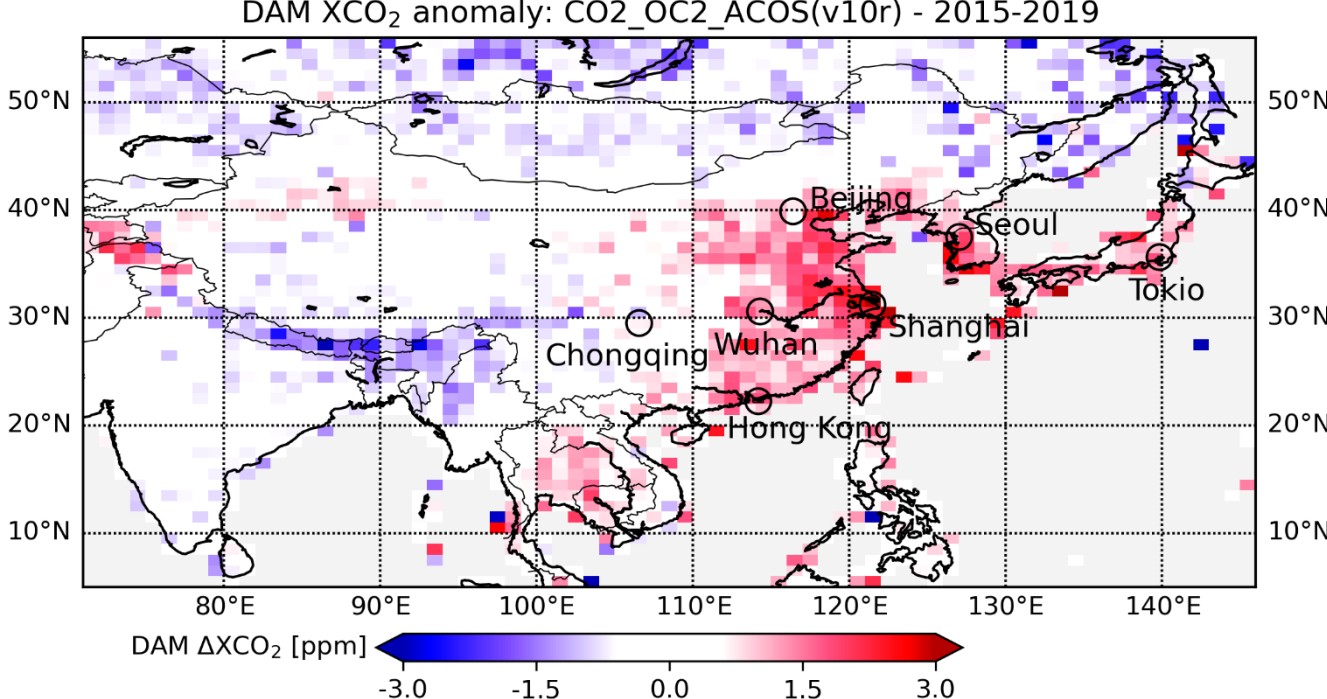

**Figure 2: As Fig. 1 but for China and surrounding areas.**






(a)

(b)

(c)

(d)

(e)

(f)

**Figure 3: As Fig. 2 but for (a) February 2015 to (f) February 2020.**




(a)

(b)

(c)

(d)

(e)

(f)

**Figure 4: Left: CT2019 XCO₂ (left, in ppm) and corresponding CO₂ surface fluxes (right, in MtCO₂/year/cell) for 15-Jan-2018**

**(first row), 15-Mar-2018 (middle) and 15-May-2018 (bottom). The red rectangle encloses the East China target region as defined**

**for this study.**



Michael.Buchwitz@iup.physik.uni-bremen.de, 20-July-2020


**Figure 5: Results obtained by applying the DAM method to CT2019 XCO₂. Top panel: The thick red line shows the CT2019 fossil fuel (FF) monthly CO₂ emissions in GtCO₂/year for target region East China. The thin grey line shows daily ΔXCO₂^FF and the blue dots show monthly ΔXCO₂^FF (see main text for a detailed explanation). Middle panel: Absolute difference between monthly ΔXCO₂^FF and the CT2019 FF target region emissions. Listed is the mean difference D, the standard deviation of the difference S, the linear**

**correlation coefficient R and the root-mean-square error (RMSE). All quantities (except R) are listed for all months (green dots) and separately for the months October to May (blue crosses). Bottom panel: the same as the middle panel but for the relative differences (as fraction, not percent, see panel title) instead of the absolute differences.**





(a)  (b)

(c)  (d)

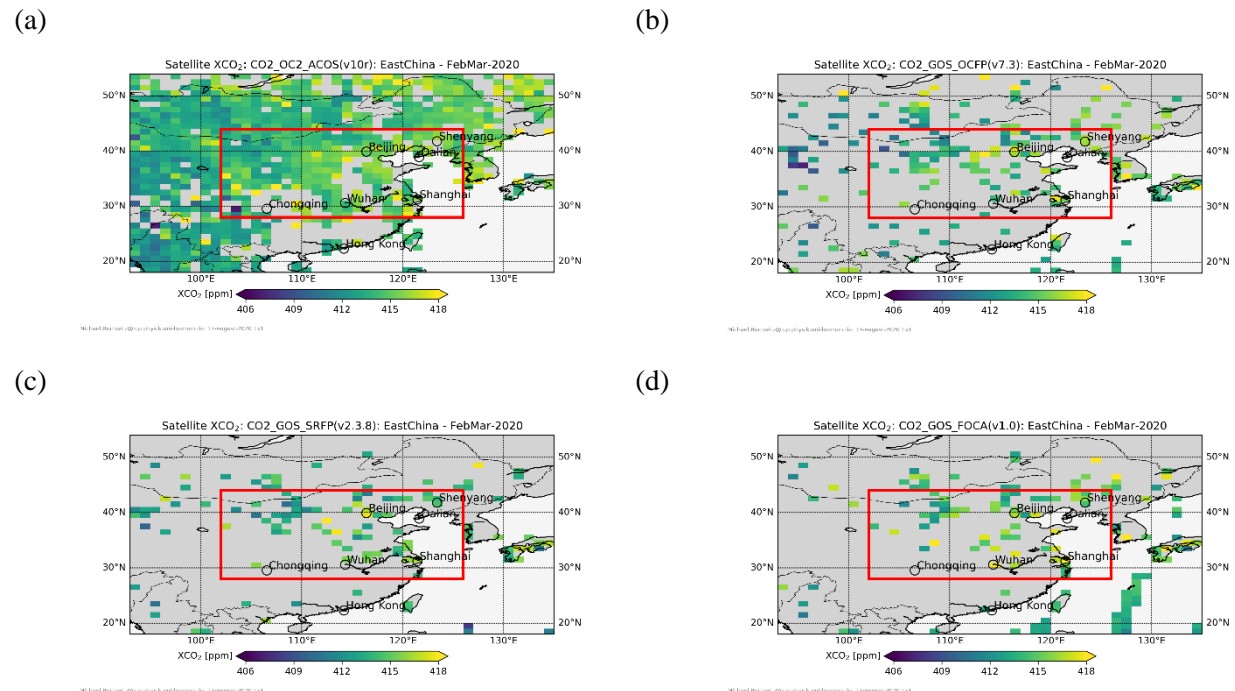


**Figure 6: (a): OCO-2 XCO₂ (version 10r, product ID CO2_OC2_ACOS) over land at 1°x1° resolution for February-March 2020. The red rectangle encloses the investigated East China target region. (b)-(d) as (a) but for products CO2_GOS_OCFP (b), CO2_GOS_SRFP (c), and CO2_GOS_FOCA (d) (see Tab. 1 for details).**




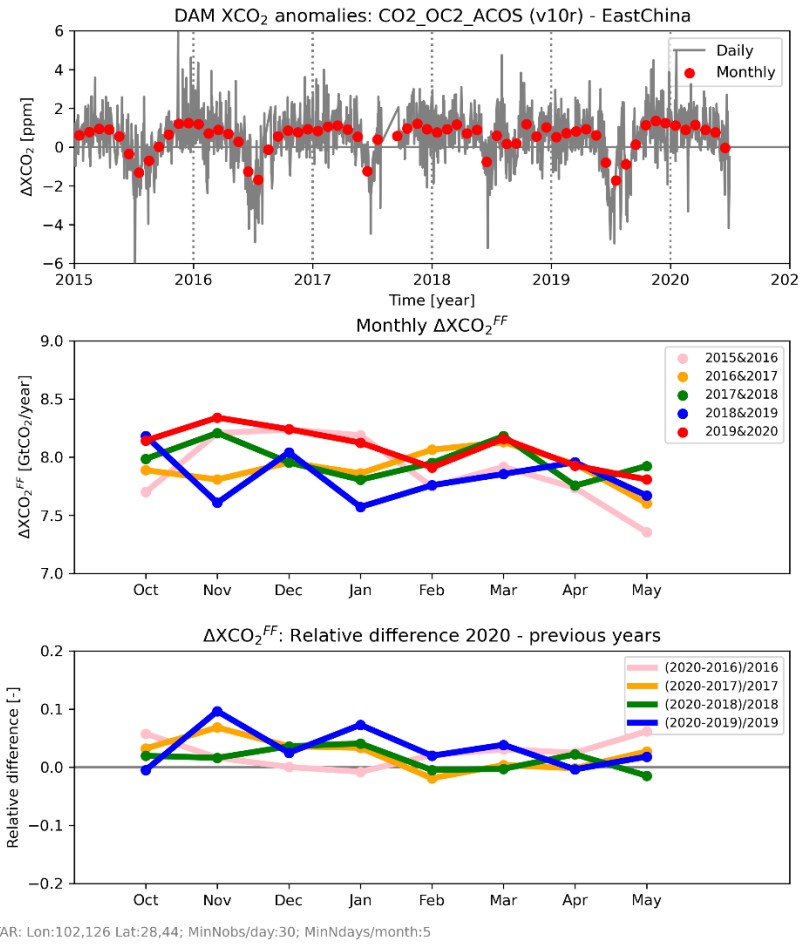

**Figure 7: DAM analysis of the OCO-2 ACOS version 10r XCO$_2$ product (CO2_OC2_ACOS) for the region East China from January 2015 to May 2020. Top: The thin grey line shows the daily DAM XCO$_2$ anomalies, i.e., daily DAM ΔXCO$_2$. The red dots are the corresponding monthly values. Middle: Monthly ΔXCO$_2^{FF}$. The red dots (and lines) refer to the time period October 2019 – May 2020, the blue dots to period October 2018 – May 2019, the green dots to period October 2017 – May 2018, etc. (see annotation). Bottom panel: the same as the middle panel but for relative differences of the monthly values: Blue dots: relative difference of the values of the red dots shown in the middle panel (ending May 2020) and the blue dots shown in the middle panel (ending May 2019) denoted in the annotation as "(2020-2019)/2019". Also shown are the relative differences for 2020 and 2018 (green), 2020 and 2017 (orange) and 2020 and 2016 (pink). The following parameters have been used to generate this figure: Minimum number of observations/day: 30, minimum number of days/month: 5.**



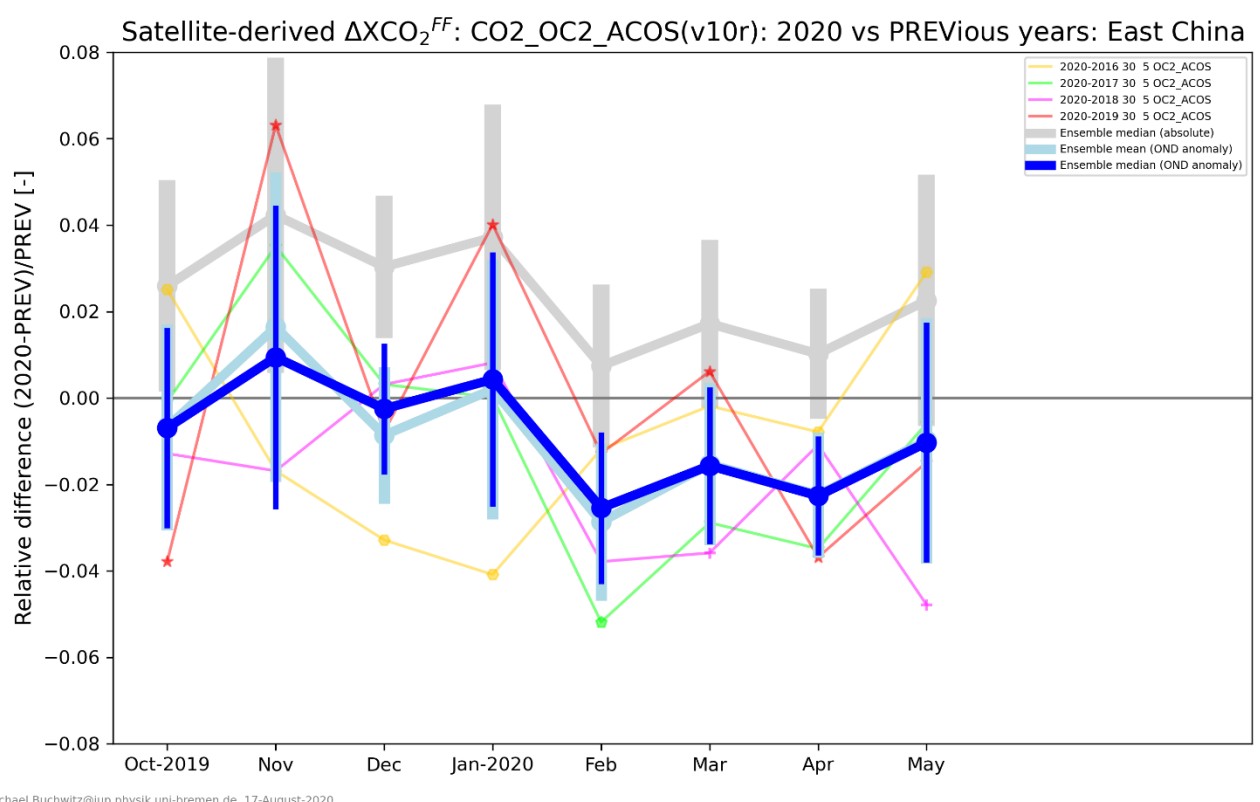


**Figure 8: Product CO2_OC2_ACOS $\Delta XCO_2^{FF}$ differences as shown in the bottom panel of Fig. 7 but including the corresponding median, mean and scatter. The relative differences as shown in the bottom panel of Fig. 7 are shown here via small symbols with thin connecting lines (using different colours for different years, see annotation) and with an offset subtracted, which corresponds to the October to December (OND) 2019 mean value, i.e., the data are shown here as anomaly relative to OND 2019 ("OND**

**anomaly"). The corresponding median and standard deviation is shown in royal blue (the corresponding mean and standard deviation is show in light blue). The median of the original data (no offset subtracted) is shown as thick grey dots and lines, i.e., the offset is the difference between the royal blue and the grey lines. The following parameters have been used to generate this figure (see also annotation): Minimum number of observations/day: 30; minimum number of days/month: 5.**




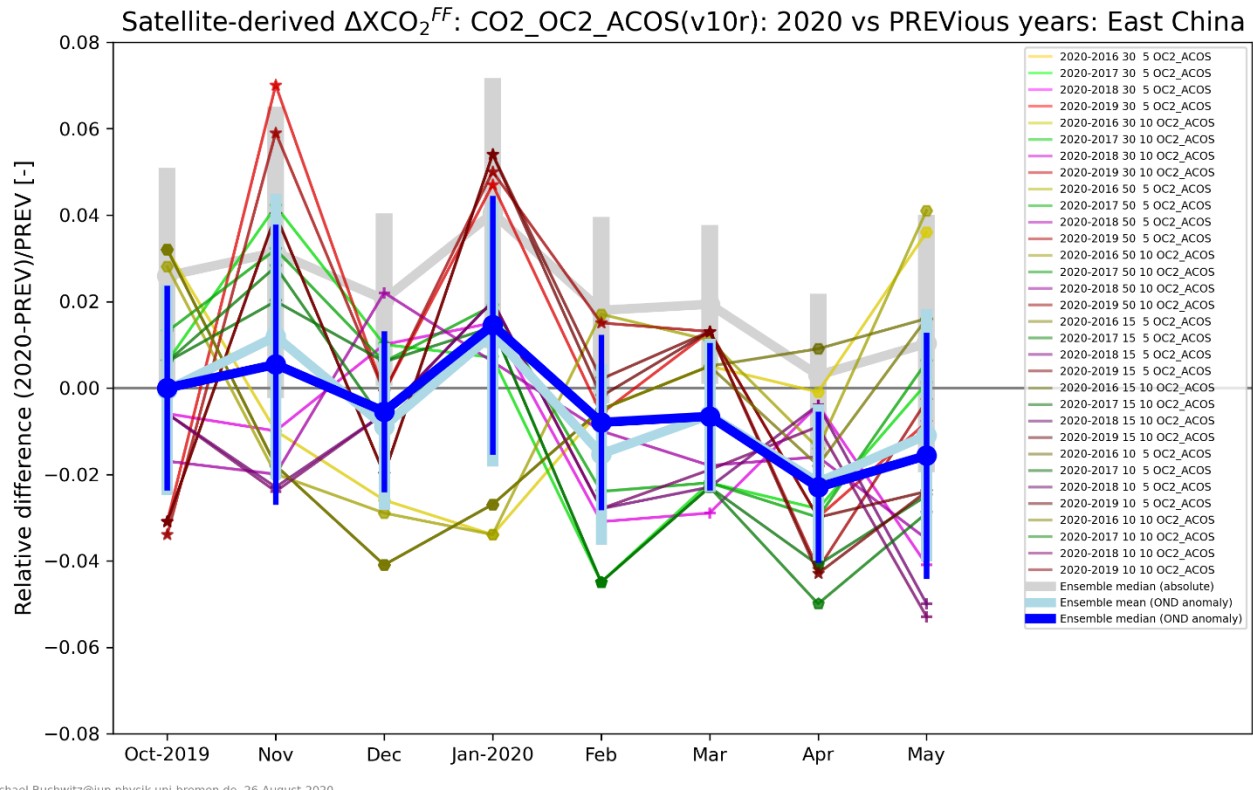

**Figure 9: The same as Fig. 8 but with additional combinations of minimum number of observations/day (30 as in Fig. 8 and in**

**addition: 50, 15 and 10) and minimum number of days/month (5 as in Fig. 8 and in addition 10) (see annotation).**





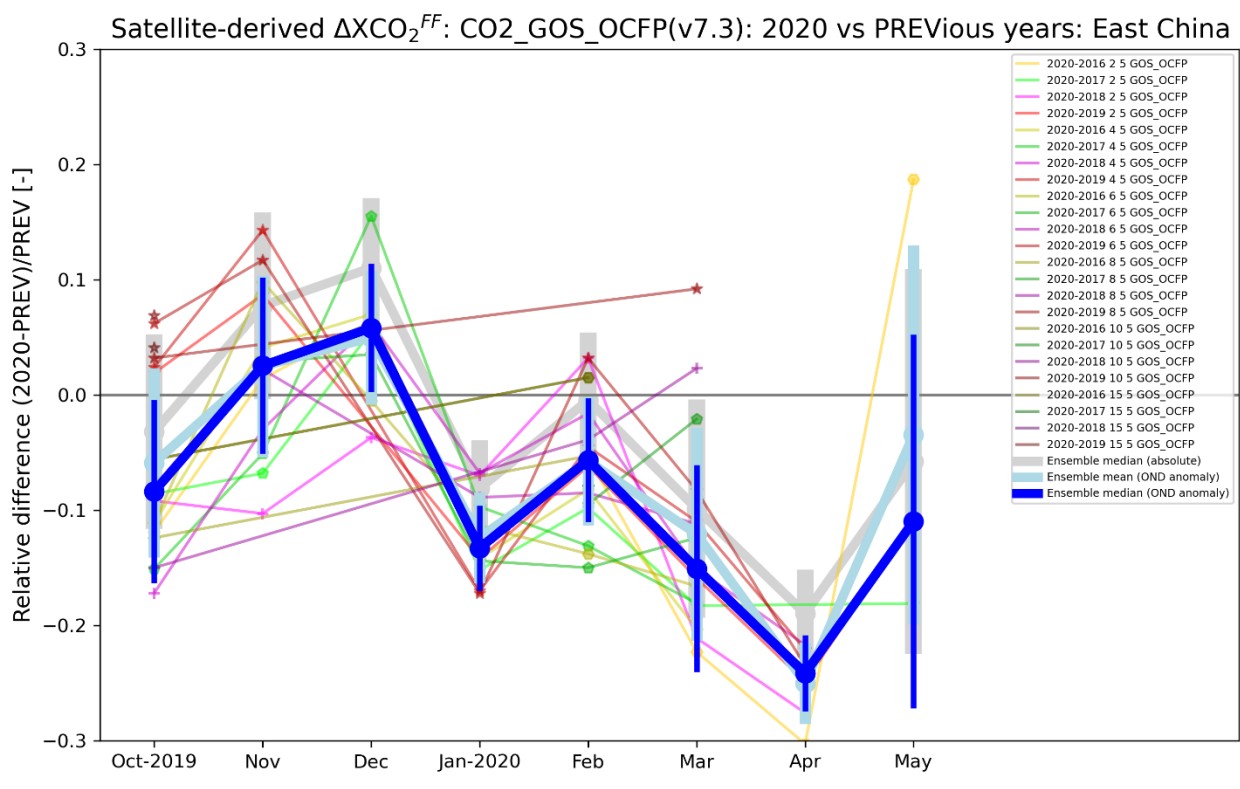


Figure 10: The same as Figs. 9 but for the product CO2_GOS_OCFP. Results are shown for several values of the required minimum number of observations/day: 2, 4, 6, 8, 10 and 15. The required minimum number of days/month is 5.






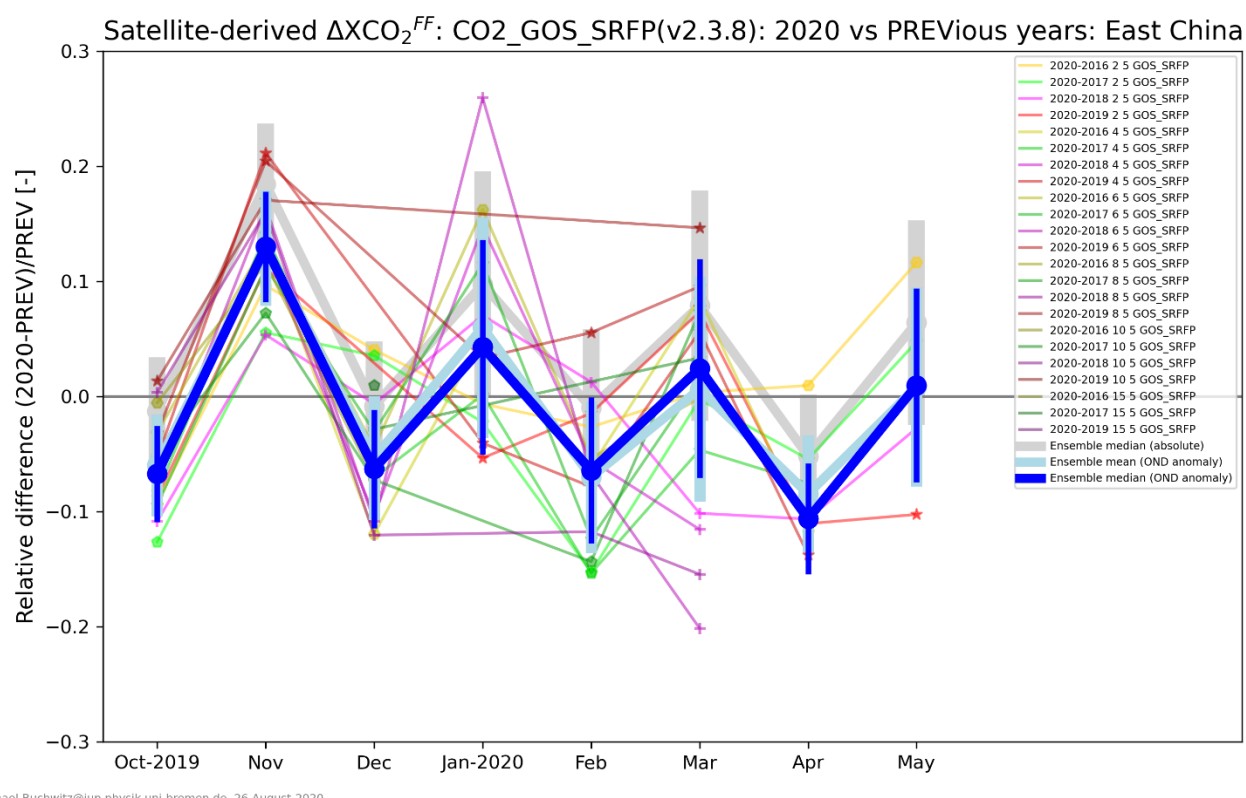

Figure 11: The same as Fig. 10 but for the product CO2_GOS_SRFP.





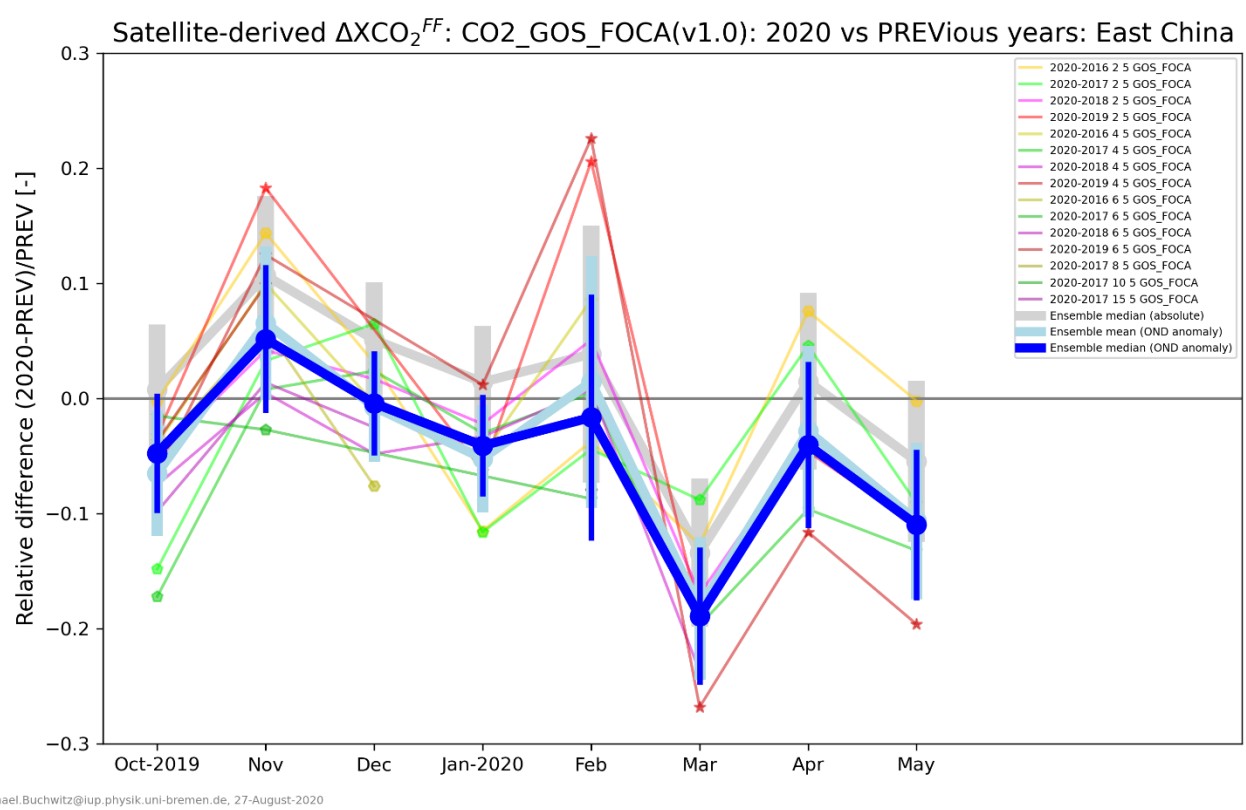


**Figure 12: The same as Fig. 10 but for the product CO2_GOS_FOCA.**

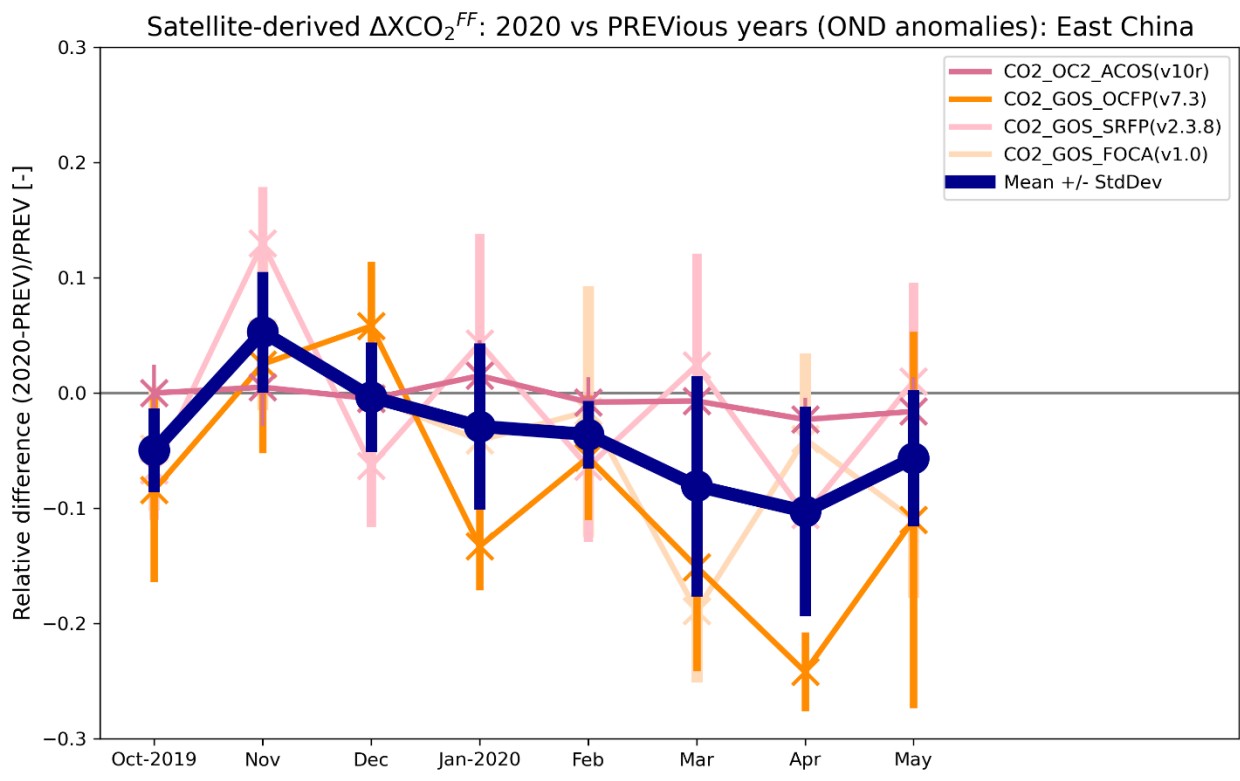

Michael.Buchwitz@iup.physik.uni-bremen.de, 9-September-2020

**Figure 13: Overview of the ensemble-based ΔXCO₂$^{FF}$ results for January-May 2020 relative to October-December 2019 and previous years (also shown in Figs. 9 – 12) via reddish colours for each of the four analysed satellite XCO₂ data products (see Tab. 1). The corresponding ensemble mean value and its uncertainty is shown in dark blue. The uncertainty has been computed as**

**standard deviation of the ensemble members. The corresponding numerical values of the ensemble members are listed in Tab. 4.**