# Peer review of "Can a regional-scale reduction of atmospheric CO2 during the COVID-19 pandemic be detected from space? A case study for East China using satellite XCO2 retrievals"

_Atmospheric Measurement Techniques, 2020_

## Referee Comment (RC1) · Anonymous Referee #1 · 5 Nov 2020

General comments The authors prepared an analysis of available satellite CO2 observations to quantify impact of CO2 emission reduction in early 2020 on the amplitude of the regional XCO2 anomaly observed over East China. The analysis is made without detailed transport modeling and thus has to rely on the magnitude of the regional mean CO2 concentration enhancements. Positive outcome of the analysis is that the change in the regional CO2 difference relative to the background was possible to detect, and the negative one was that the uncertainty appears to be of the same order as signal. Authors cite weak signal, large variability in observation/cloud coverage and

impact from biospheric fluxes on XCO2 as complicating factors. The elaborate analysis is a valuable addition to a body of evidence on capability of global carbon observing system to detect the short and long-term changes in CO2 emissions and sinks. The paper is well written and can be published after applying minor revisions and technical corrections.

Detailed comments

Introduction. Authors can use opportunity to mention more recent publications on the topic, complimentary to this study (Chevallier et al. 2020; Tohjima et al. 2020; Zeng et al. 2020)

L205-210 Not clear if the CarbonTracker-derived scaling of XCO2 to FF emissions helps correcting for year to year variability in wind speed, or it is a climatology. Need to clarify. It can be mentioned Zheng et al, 2020b used transport model for a similar purpose.

L336 The DAM method is not supposed to extract exclusively anthropogenic emission contributions to XCO2, it is better to revise the sentence accordingly.

L355-362 The discussion gives impression that the satellite observation/retrieval errors are most important, while the contribution of the short and long-range transport including both biogenic and fossil signals is not explicitly mentioned, while it is likely to contribute to differences between different time periods, especially across GOSAT products

L365 If errors do not scale with the inverse of the square root of number of observations, then those may be not random enough.

Technical corrections

L92 Suggest checking reference format: Sussmann and Rettinger, 2020, or (2020) L770 Zheng et al. paper status changed to published.

References

Chevallier, F., Zheng, B., Broquet, G., Ciais, P., Liu, Z., Davis, S. J., et al. Local anomalies in the column‐averaged dry air mole fractions of carbon dioxide across the globe during the first months of the coronavirus recession. Geophysical Research Letters, 47, e2020GL090244, https://doi.org/10.1029/2020GL090244, 2020.

Tohjima, Y., Patra, P.K., Niwa, Y. et al. Detection of fossil-fuel CO2 plummet in China due to COVID-19 by observation at Hateruma. Sci Rep 10, 18688. https://doi.org/10.1038/s41598-020-75763-6, 2020.

Zeng N., Han P., Liu D., Liu Z., Oda T., Martin C., Liu Z., Yao B., Sun W., Wang P., Cai Q., Dickerson R., Maksyutov S. Global to local impacts on atmospheric CO2 caused by COVID-19 lockdown. https://arxiv.org/abs/2010.13025, 2020

---

## Referee Comment (RC2) · Anonymous Referee #2 · 19 Dec 2020

This paper endeavors to characterize the impacts of COVID-19 pandemic on atmospheric  $CO_2$  by estimating the fossil fuel emission from satellite observations (OCO-2 and GOSAT). This is an inverse estimation, so a model is needed to establish the relation between observation and model variables (Fossil fuel emissions (FF)). The authors did not use a physical model, but use the posterior  $CO_2$  field and input fossil fuel emissions from CarbonTracker – a inverse model of atmospheric  $CO_2$ , to construct a linear regression model, to calculate FF emissions from the change of  $XCO_2$ , and this relationship was then used to estimate emissions from satellite  $XCO_2$  observations. The

authors did get an estimation of a small change in fossil fuel emissions, but the change is so small comparing to its uncertainty and possible variations caused by other factors.

Even so, this referee suggests that manuscript should be published after major revision, as suggested below.

Major comments:

1. The paper is too long comparing to its contents. For example, the lengthy abstract, and a couple of paragraphs (part of) quoted from other documents, and other redundant description and analysis.

2. Reduction algorithm is the core of the method using in the paper, and in the same time the authors did not get a significant change as a result of COVID-19 pandemic. Therefore, if we need to justify the result of this research, the authors should assess the consequence of a possible signal lost of the original observations as a result of the reduction algorithm used, and this could be the most important contribution of this paper to our research community.

Minor comments:

The abstract is way too long.

Line 138-139: "assimilates...as well as...". Does the model assimilate emissions?

Line 142-149: Is it necessary to quote a whole paragraph to describe CT?

Line 154: "The DAM method is essentially identical with the" and Line 159: "Our approach is very similar". If you think "essentially identical" and "very similar" are identical, then "very similar" in Line 159 is redundant.

Line 156-159: Hakkarainen et al., 2019, explain their method as follows: "...". Is it necessary?

Line 163: How about change "but" to "and"? you already have a "but" in line 162.

AMTD
Line 164: how about remove "as contained"?

Line 167: What is "very similar"?

Line 168: what is "The good agreement"?

Line 208:  $\Delta XCO_2^{FF}$  is misleading. It is FF estimated from  $\Delta XCO_2$ , and  $FF^{\Delta XCO_2}$  could be more intuitive.

Line 361: "This single observations uncertainty". Is "This single uncertainty of observations" better?

---

## Referee Comment (RC3) · Anonymous Referee #3 · 17 Jan 2021

The manuscript "Can a regional-scale reduction of atmospheric CO2 during the COVID-19 pandemic be detected from space? A case study for East China using satellite XCO2 retrievals" attempts at detecting a significant anomaly of Fossil Fuel (FF) CO2 emissions in East China due to the impact of the covid-19 crisis from available satellite XCO2 data. The study proposes a relatively simple approach to derive monthly FFCO2 emission from averages of spatial anomalies in XCO2 data.

It properly assesses the significance of the derived changes in emissions at the beginning of 2020 by comparing results from different XCO2 datasets and by analyzing

the temporal variability of the estimates of monthly emissions over different years. It raises careful conclusions regarding the current potential to detect large decreases in the emissions such as that caused by the covid-19 crisis. This study should contribute to the understanding of the current capabilities and needs for the monitoring of the CO2 emissions from space.

However, I have several major concerns regarding the analysis and I think that they should be addressed or at least discussed properly before the study can be published.

1) The main one is about the linear regression used to convert averages of spatial anomalies of XCO2 into monthly FF CO2 emissions. The result of the regression (p7 l209) shows an offset (7.1GtCO2/year) which is close to the total emissions of the studied area (p6, l181). Therefore, the term proportional to the spatial XCO2 anomalies is much smaller. Assuming that natural fluxes (and other types of anthropogenic emissions) are null during the period of analysis, the spatial anomalies of XCO2 should be proportional to the FF emissions in the area. The explanations for getting an offset should be, in principle, perturbing factors such as some atmospheric transport effect or the other types of fluxes. However, I feel that these could hardly explain such a large offset dominating the total FF CO2 emissions estimates. I may have misunderstood something, but I think that the authors need to provide some explanations for this large offset. Actually, lines 212-221 p7 stress rather than explain the problem. If this offset challenges any physical explanation for the linear regression, the authors should justify why the linear regression could provide meaningful results.

2) I hardly understand how the derivation of spatial anomalies of XCO2 based on the crude extraction of median values over latitudinal bands could provide useful quantifications. The complex spatial distribution of fluxes, the complex combination between fluxes and atmospheric transport and the complex distribution of satellite data in space and time (far from homogeneous) makes it hardly meaningful physically, unless, again, I have misunderstood a critical point regarding this computation. "The background" is a misleading term which do not really mean anything by itself. Line 155 p5: which

"trends and seasonal variations" do they want to remove ? why would it isolate "CO2 source/sink signals" ? I do not find any real rationale behind p5 l158-159 ("the median was chosen because..."). I assume that such anomalies can be highly misleading as strongly suggested by figures 1 and 2. Should we read from line 27-28 p1 or line 172-173 p6 together with these figures that some of highest sinks of CO2 occur in Himalayas and Sahara ? and that there is not any large sink region within North America ? I would be curious to see where are the locations corresponding to the median values per latitudinal band, and in particular the comparisons between these locations in CT2019 and in the XCO2 data. A computation of the anomalies of XCO2 over East China against upwind areas or at least against concentrations in neighbour regions could probably make more sense. Could the offset arising from the linear regression (see (1)) be a consequence of this computation of the anomalies ? The fact that previous publications have used such computations does not appear as a sufficient justification to me. I think that the authors should better support the suitability of this approach for the analysis conducted here. Given its simplicity, its lack of theoretical basis and its weaknesses, does this computation really deserve a label such as "DAM" which suggests that it is a well proven standard approach for analyzing XCO2 data ?

=> (1) and (2) feed the conclusion raised by the authors themselves that their method could have been too simple. This could appear as the main reason for the lack of ability to extract a clear signal from the covid-19 crisis. The authors need to have and share a stronger confidence in the relevance of these computations.

3) a) Even though it is mentioned from time to time in the discussion (it often comes too late), the spatial and temporal scale of analysis is not really properly characterized, justified, and discussed while it has critical consequences on the outcomes and conclusions of the study. The introduction provides numbers at scales that are not really defined, probably mixing different scales e.g. at lines 100-102 and l110 p4. Comparisons between the FFCO2 signal "at regional scale" and the noise on individual XCO2 data hardly makes sense (e.g. l103-104 p4 and 358-360 p12). Stating that averaging

is needed to decrease the random noise (line 105 p4) is surprising: there is no need to average data over the whole East China and 1 month to ensure the strong reduction of the noise, and appropriate analysis at fine scales should denoise the data much better than simple averaging. The large scale ("regional") XCO2 fields are often dominated by the signal from CO2 natural fluxes and the signal from FFCO2 emissions is generally sought at fine spatial scales. Enlarging the scales of analysis can bury the signal from FFCO2 emissions in that of natural fluxes. One could have negative conclusions with "regional scale" analysis and positive ones from analysis at finer scales as acknowledge a bit late by the last lines of the manuscript. b) Even if East China has a relatively high large scale signal from FFCO2 emissions and even if the analysis are focused on the late fall - early spring period, figure 4 strongly suggests that large patterns from natural fluxes overlap the area of analysis, which could strongly bias the linear regression. Biogenic fluxes and anthropogenic emissions follow a seasonal cycle with a very similar phase and thus the regression may be able to fit the general variations of FFCO2 emissions by assimilating data with a strong signal from the biogenic fluxes. A better care for such a source of uncertainty needs to be taken.

4) The authors claim several times that their method is "data driven" and does not rely on a priori knowledges about the fluxes and atmospheric transport. This is contradicted by the use of the CT2019 system to derive the linear regression between XCO2 spatial anomalies and FFCO2 emissions. This is a critical point: the authors do need to simulate the link between emissions and concentrations using a proxy for the atmospheric transport and some assumptions regarding the spatial distribution of the FF emissions and the other fluxes. Rather than ignoring the atmospheric transport and a priori knowledges on fluxes, they use a crude proxy for it, which could bring large uncertainties to the analysis.

5) In principle, the coarse spatial resolution of CT2019 could be a problem for the analysis of the signal from FFCO2 emissions in XCO2 data which have much finer spatial footprints. How are the XCO2 values derived from this model ? When applying the

"DAM" to CT2019 and then deriving the linear regression from this system, is CT2019 subsampled at the satellite locations ? if yes, why would there be one equation only for the linear regression while the authors apply it to four satellite datasets with different samplings ? if not, can the linear regression from full emission and concentration fields really apply to the satellite data, which have a sparse spatial sampling ?

6) Some parts of the text have been written too quickly (with some lack of clarity, repetitions, copy paste of paragraphs from other publications to describe the model or methods, some typos...). The main text should not detail the color code of the figures (the figures should be clear enough). I think that there are too many figures inserted in the main text. Some could be moved to supplementary material. Others could be merged together into synthetic figures.

Secondary points:

- the authors indicate that this study is the "first attempt to determine whether a regional-scale reduction of anthropogenic CO2 emissions during the COVID-19 pandemic can be detected using space-based observations of atmospheric CO2" (p1 l20) -> given the speed of some scientific studies and the rapid publication of papers, especially during the covid-19 crisis, this claim may not hold. It may depend on the precise definition for the term "regional scale" used here. In any case, such a fact should not lower the need for strong justifications for the simple and quick analysis methods applied here (see (1) and (2)).

- a source of misunderstanding when discussing increase of decrease of emissions due to covid-19 is whether comparisons apply to previous months or to identical calendar month of the previous year(s). Things are even more complex here where changes from year-1 to current year of changes from month-1 to current month of emissions are analyzed. Even though it may hamper the fluence of the text, the description of the anomalies should systematically be clear regarding this if the context of the sentence does not help (e.g. line 42 on p2, line 83 p3, line 352 p12).

- Should we understand from this paper that the anomaly of FFCO2 emissions due to covid-19 can be quantified from the TCCON network (line 94 p3) but not from the XCO2 GOSAT and OCO-2 data even though the TCCON network is too sparse to assess the satellite data accuracy at regional scale in East China (end of introduction) ?

- The "complementarity" between OCO-2 and GOSAT (line 126 p4) could be further discussed.  So far, I feel that there has not been much expectations regarding the ability of GOSAT to allow for the quantification of FFCO2 emissions, while various studies have attempted at quantifying FFCO2 emissions based on the OCO-2 data.

- l162 to 169 p5-6 display a loose discussion on whether the width of latitudinal bands is important or not, with some opposition between l163 vs. l162 and 168-169. The "good agreement" discussed on line 169 sounds qualitative while the anomalies will be used for the quantification of emission temporal variations in East China. Some quantitative assessment of the sensitivity to the width of latitudinal band would have made more sense.

- isn't the notation "deltaXCO2FF" for the estimate of the emissions misleading ? this is an estimate of the total FF emissions, not of the FF component of the XCO2 anomalies.

- table 1: should not the labels "generated by the authors (#)" be changed ? table 3 is useless, the coordinates should just be given in the text.

---

## Author Comment (AC1) · 10 Feb 2021

Many thanks for taking the time to review our manuscript and for providing very useful feedback. Your comments and the comments from the other two referees have been carefully taken into account when generating the revised version of our manuscript. Please see below our response to each of your comments.

Referee: General comments The authors prepared an analysis of available satellite CO2 observations to quantify impact of CO2 emission reduction in early 2020 on the

amplitude of the regional XCO2 anomaly observed over East China. The analysis is made without detailed transport modeling and thus has to rely on the magnitude of the regional mean CO2 concentration enhancements. Positive outcome of the analysis is that the change in the regional CO2 difference relative to the background was possible to detect, and the negative one was that the uncertainty appears to be of the same order as signal. Authors cite weak signal, large variability in observation/cloud coverage and impact from biospheric fluxes on XCO2 as complicating factors. The elaborate analysis is a valuable addition to a body of evidence on capability of global carbon observing system to detect the short and long-term changes in CO2 emissions and sinks. The paper is well written and can be published after applying minor revisions and technical corrections.

Detailed comments

Referee: Introduction. Authors can use opportunity to mention more recent publications on the topic, complimentary to this study (Chevallier et al. 2020; Tohjima et al. 2020; Zeng et al. 2020)

Author's response: We have added references to these publications for the revised version of our manuscript.

Referee: L205-210 Not clear if the CarbonTracker-derived scaling of XCO2 to FF emissions helps correcting for year to year variability in wind speed, or it is a climatology. Need to clarify. It can be mentioned Zheng et al, 2020b used transport model for a similar purpose.

Author's response: CarbonTracker data until end of 2018 were available when we started carrying out our study and for this study we used the last four years (2015-2018). During this time period we did not observe any time dependency such as a trend or anomalous years when looking at scaled XCO2 anomalies versus FF emissions (see our Fig. 5). However, it cannot be ruled out that 2019 or 2020 were significantly different compared to previous years with respect to aspects relevant for our

study. We essentially assume that this is not the case or that unconsidered variability is captured by our uncertainty estimates, which are based on differences of the October 2019 to May 2020 period and previous October to May periods. We improved related explanations in the revised version of our paper. We prefer not to refer to Zheng et al, 2020b, in this context as they used a different method and because this would require to also cite and summarize several other publications which would lengthen the paper with probably only limited added value.

Referee: L336 The DAM method is not supposed to extract exclusively anthropogenic emission contributions to XCO2, it is better to revise the sentence accordingly.

Author's response: Agreed. We will add the following: "(note however that the FMI method is not supposed to extract exclusively anthropogenic emission contributions to XCO2, see Hakkarainen et al., 2019)".

Referee: L355-362 The discussion gives impression that the satellite observation/retrieval errors are most important, while the contribution of the short and long-range transport including both biogenic and fossil signals is not explicitly mentioned, while it is likely to contribute to differences between different time periods, especially across GOSAT products.

Author's response: Agreed. To consider this, we have added the following: "Of course also other sources of uncertainty are relevant in this context, in particular time dependent atmospheric transport and varying biogenic CO2 contributions (e.g., Houweling et al., 2015, and references given therein).".

Referee: L365 If errors do not scale with the inverse of the square root of number of observations, then those may be not random enough.

Author's response: Yes, this is true. There are (unknown) systematic errors and error correlations. We have added this information. Technical corrections

Referee: L92 Suggest checking reference format: Sussmann and Rettinger, 2020, or

(2020)

Author's response: Many thanks. We have harmonized the reference format.

Referee: L770 Zheng et al. paper status changed to published.

Author's response: We have updated this.

References

Chevallier, F., Zheng, B., Broquet, G., Ciais, P., Liu, Z., Davis, S. J., et al. Local anomalies in the column-averaged dry air mole fractions of carbon dioxide across the globe during the first months of the coronavirus recession. Geophysical Research Letters, 47, e2020GL090244, https://doi.org/10.1029/2020GL090244, 2020.

Houweling, S., Baker, D., Basu, S., Boesch, H., Butz, A., Chevallier, F., Deng, F., Dlugokencky, E. J., Feng, L., Ganshin, A., Hasekamp, O., Jones, D., Maksyutov, S., Marshall, J., Oda, T., O'Dell, C. W., Oshchepkov, S., Palmer, P. I., Peylin, P., Poussi, Z., Reum, F., Takagi, H., Yoshida, Y., and Zhuralev, R.: An intercomparison of inverse models for estimating sources and sinks of CO2 using GOSAT measurements, J. Geophys. Res. Atmos., 120, 5253–5266, doi:10.1002/2014JD022962, 2015.

Tohjima, Y., Patra, P.K., Niwa, Y. et al. Detection of fossil-fuel CO2 plummet in China due to COVID-19 by observation at Hateruma. Sci Rep 10, 18688. https://doi.org/10.1038/s41598-020-75763-6, 2020. Zeng N., Han P., Liu D., Liu Z., Oda T., Martin C., Liu Z., Yao B., Sun W., Wang P., Cai Q., Dickerson R., Maksyutov S. Global to local impacts on atmospheric CO2 caused by COVID-19 lockdown. https://arxiv.org/abs/2010.13025, 2020.

---

## Author Comment (AC2) · 10 Feb 2021

Many thanks for taking the time to review our manuscript and for providing very useful feedback. Your comments and the comments from the other two referees have been carefully taken into account when generating the revised version of our manuscript. Please see below our response to each of your comments.

Referee: This paper endeavors to characterize the impacts of COVID-19 pandemic on atmospheric CO2 by estimating the fossil fuel emission from satellite observations

(OCO-2 and GOSAT). This is an inverse estimation, so a model is needed to establish the relation between observation and model variables (Fossil fuel emissions (FF)). The authors did not use a physical model, but use the posterior CO2 field and input fossil fuel emissions from CarbonTracker – a inverse model of atmospheric CO2, to construct a linear regression model, to calculate FF emissions from the change of XCO2, and this relationship was then used to estimate emissions from satellite XCO2 observations. The authors did get an estimation of a small change in fossil fuel emissions, but the change is so small comparing to its uncertainty and possible variations caused by other factors. Even so, this referee suggests that manuscript should be published after major revision, as suggested below.

Major comments:

Referee: 1. The paper is too long comparing to its contents. For example, the lengthy abstract, and a couple of paragraphs (part of) quoted from other documents, and other redundant description and analysis.

Author's response: We have carefully checked the paper for unnecessary redundancy and have shortened the paper where possible. We have significantly shortened the abstract and have removed the quotes from other documents. However, we think that a certain level of redundancy helps readers to more easily and faster understand what has been done and what has been concluded from this and why. We have aimed at reducing repetition to a minimum while at the same time writing the paper such that it is easy to read and understand without the need go frequently back and forth. Furthermore, we have improved the structure of our paper to also better meet this goal.

Referee: 2. Reduction algorithm is the core of the method using in the paper, and in the same time the authors did not get a significant change as a result of COVID-19 pandemic. Therefore, if we need to justify the result of this research, the authors should assess the consequence of a possible signal lost of the original observations as a result of the reduction algorithm used, and this could be the most important contribution of

this paper to our research community.

Author's response: It would quite challenging (if not impossible) to reliably quantify a possible information loss due to our admittedly quite simple analysis method. One way to assess this could be to use one or several more traditional inverse modelling method including detailed transport modelling and use of a priori information on CO2 surface fluxes. This however would be a major exercise including detailed transport modelling etc. (which we consider out of scope of our study) and also would not guarantee that significantly more information can be extracted. There are several reasons for this including transport modelling errors, uncertainties of a priori fluxes (fossil fuel, biogenic and other), the need to consider (unknown or not well enough known) error correlations of the satellite retrievals, etc. Ideally, when trying to uncover a tiny signal, one should attempt to remove all other competing/confounding signals. We use two methods, but these are not the only possibilities and there may be other methods to more effectively remove the confounding signals of biology, transport, satellite sampling, etc. After submission of our manuscript other publications appeared using different approaches to find out to what extent satellite XCO2 retrievals can provide COVID-19 related CO2 emission reduction information (see Chevallier et al., 2020, Tohjima et al., 2020, and Zeng et al., 2020, now also cited in the revised version of our manuscript). The findings of these studies are consistent with the conclusions drawn in our manuscript. This is not a proof that the limit has already been achieved via our analysis, but this is a strong indication that a significant information loss due to the simple data-driven analysis method used in our publication is not very likely but also not entirely impossible.

Minor comments:

Referee: The abstract is way too long.

Author's response: Agreed. The abstract has been considerably shortened, see also above.

Referee: Line 138-139: "assimilates: : :as well as: : :". Does the model assimilate

emissions? Line 142-149: Is it necessary to quote a whole paragraph to describe CT?

Author's response: We have revised the entire paragraph along the lines suggested.

Referee: Line 154: "The DAM method is essentially identical with the" and Line 159: "Our approach is very similar". If you think "essentially identical" and "very similar" are identical, then "very similar" in Line 159 is redundant.

Author's response: We have revised the sentence by removing "very similar".

Referee: Line 156-159: Hakkarainen et al., 2019, explain their method as follows: ": : :". Is it necessary?

Author's response: We have removed the quote as suggested.

Referee: Line 163: How about change "but" to "and"? you already have a "but" in line 162.

Author's response: Agreed. We will improve the sentence by splitting it into two: "Our investigations showed that the width of the latitude band is not critical. The band needs to be wide enough to contain a statistically significant sample, but narrow enough to resolve large latitudinal gradients in CO2.".

Referee: Line 164: how about remove "as contained"?

Author's response: Yes, we removed this.

Referee: Line 168: what is "The good agreement"?

Author's response: We will replace "The good agreement confirms" with "The degree of agreement confirms".

Referee: Line 208: $\Delta$XCO2FF is misleading. It is FF estimated from $\Delta$XCO2, and FF$\Delta$XCO2 could be more intuitive.

Author's response: Agreed. We will change the notation.

Referee: Line 361: "This single observations uncertainty". Is "This single uncertainty of observations" better?

Author's response: What we mean is the uncertainty of single observations (rather than the uncertainty of averaged observations). We will write: "The uncertainty of single observations, which is typically around …".

References

Chevallier, F., Zheng, B., Broquet, G., Ciais, P., Liu, Z., Davis, S. J., et al. Local anomalies in the column-averaged dry air mole fractions of carbon dioxide across the globe during the first months of the coronavirus recession. Geophysical Research Letters, 47, e2020GL090244, https://doi.org/10.1029/2020GL090244, 2020.

Tohjima, Y., Patra, P.K., Niwa, Y. et al. Detection of fossil-fuel CO2 plummet in China due to COVID-19 by observation at Hateruma. Sci Rep 10, 18688. https://doi.org/10.1038/s41598-020-75763-6, 2020.

Zeng N., Han P., Liu D., Liu Z., Oda T., Martin C., Liu Z., Yao B., Sun W., Wang P., Cai Q., Dickerson R., Maksyutov S. Global to local impacts on atmospheric CO2 caused by COVID-19 lockdown. https://arxiv.org/abs/2010.13025, 2020.
* * *

---

## Author Comment (AC3) · 10 Feb 2021

Many thanks for taking the time to review our manuscript and for providing very useful feedback. Your comments and the comments from the other two referees have been carefully taken into account when generating the revised version of our manuscript. Please see below our response to each of your comments.

Referee: The manuscript "Can a regional-scale reduction of atmospheric CO2 during the COVID-19 pandemic be detected from space? A case study for East China using satellite XCO2 retrievals" attempts at detecting a significant anomaly of Fossil Fuel (FF) CO2 emissions in East China due to the impact of the covid-19 crisis from available satellite XCO2 data. The study proposes a relatively simple approach to derive monthly FFCO2 emission from averages of spatial anomalies in XCO2 data. It properly assesses the significance of the derived changes in emissions at the beginning of 2020 by comparing results from different XCO2 datasets and by analysing the temporal variability of the estimates of monthly emissions over different years. It raises careful conclusions regarding the current potential to detect large decreases in the emissions such as that caused by the covid-19 crisis. This study should contribute to the understanding of the current capabilities and needs for the monitoring of the CO2 emissions from space. However, I have several major concerns regarding the analysis and I think that they should be addressed or at least discussed properly before the study can be published.

Author's response: For the revised version of our manuscript we did our best to consider all your comments, and our detailed point-by-point response is given in following.

Referee: 1) The main one is about the linear regression used to convert averages of spatial anomalies of XCO2 into monthly FF CO2 emissions. The result of the regression (p7 l209) shows an offset (7.1GtCO2/year) which is close to the total emissions of the studied area (p6, l181). Therefore, the term proportional to the spatial XCO2 anomalies is much smaller. Assuming that natural fluxes (and other types of anthropogenic emissions) are null during the period of analysis, the spatial anomalies of XCO2 should be proportional to the FF emissions in the area. The explanations for getting an offset should be, in principle, perturbing factors such as some atmospheric transport effect or the other types of fluxes. However, I feel that these could hardly explain such a large offset dominating the total FF CO2 emissions estimates. I may have misunderstood something, but I think that the authors need to provide some explanations for this large offset. Actually, lines 212-221 p7 stress rather than explain the problem. If this offset challenges any physical explanation for the linear regression, the

authors should justify why the linear regression could provide meaningful results.

Author's response: We agree that this requires additional explanations. You are right that "Assuming that natural fluxes (and other types of anthropogenic emissions) are null during the period of analysis, the spatial anomalies of XCO2 should be proportional to the FF emissions in the area.". The problem is in fact related to the biogenic fluxes. To better address this, we will replace the relevant figure (Fig. 5) with an improved one which also shows other relevant quantities contained in the CT2019 data set, especially we will add XCO2 anomalies due to fossil fuel (FF) and biogenic (BIO) fluxes separately, and we will also modify the text significantly. The reason for the large offset is the influence of the biosphere. We will show in the revised version of the paper that during October to May the XCO2 anomalies are mostly dominated by FF emissions but of course the BIO fluxes are not zero. Especially around May the uptake of atmospheric CO2 due to the biosphere is so large that the XCO2 anomalies due to BIO fluxes are on the same order (but with an opposite sign) as the XCO2 anomalies due to FF emissions (which are always positive), i.e., the XCO2 anomalies can be close to zero despite positive FF emissions (or the FF related XCO2 anomalies). To deal with this, the offset is needed. Of course our method is not perfect and we have to quantify the error of our method (and we do this). Also this aspect will be improved for the revised version of our paper.

Referee: 2) I hardly understand how the derivation of spatial anomalies of XCO2 based on the crude extraction of median values over latitudinal bands could provide useful quantifications. The complex spatial distribution of fluxes, the complex combination between fluxes and atmospheric transport and the complex distribution of satellite data in space and time (far from homogeneous) makes it hardly meaningful physically, unless, again, I have misunderstood a critical point regarding this computation. "The background" is a misleading term which do not really mean anything by itself. Line 155 p5: which "trends and seasonal variations" do they want to remove ? why would it isolate "CO2 source/sink signals" ? I do not find any real rationale behind p5 l158-159 ("the

median was chosen because..."). I assume that such anomalies can be highly misleading as strongly suggested by figures 1 and 2. Should we read from line 27-28 p1 or line 172-173 p6 together with these figures that some of highest sinks of CO2 occur in Himalayas and Sahara ? and that there is not any large sink region within North America ? I would be curious to see where are the locations corresponding to the median values per latitudinal band, and in particular the comparisons between these locations in CT2019 and in the XCO2 data. A computation of the anomalies of XCO2 over East China against upwind areas or at least against concentrations in neighbour regions could probably make more sense. Could the offset arising from the linear regression (see (1)) be a consequence of this computation of the anomalies ? The fact that previous publications have used such computations does not appear as a sufficient justification to me. I think that the authors should better support the suitability of this approach for the analysis conducted here. Given its simplicity, its lack of theoretical basis and its weaknesses, does this computation really deserve a label such as "DAM" which suggests that it is a well proven standard approach for analyzing XCO2 data ?

Author's response: We have defined in our paper how "The background" is computed and how it is used. We therefore cannot see why this is "a misleading term which do not really mean anything by itself". Nevertheless, for the revised version of the paper, we will add additional information to further improve this aspect. Concerning "I hardly understand how the derivation of spatial anomalies of XCO2 based on the crude extraction of median values over latitudinal bands could provide useful quantifications.": Hakkarainen et al., 2019, show detailed comparisons with model simulations (from multi-year averages to seasonal maps etc.) and we see no need to repeat this. However, the comparisons reported in Hakkarainen et al., 2019, are mostly qualitative. Therefore, in our publication, we have investigated to what extent the resulting XCO2 anomalies can be used quantitatively to obtain emission estimates. For this purpose we used the CT2019 model data set and processed this data set in the same way as we process the satellite data. Concerning your proposal "A computation of the anomalies of XCO2 over East China against upwind areas or at least against concentrations

in neighbour regions could probably make more sense.": Computations of upwind area anomalies is an option but would require an entirely different method including addressing the questions how to define upwind area anomalies, etc. But to address your second proposal we will add results from a second method, not based on latitude band medians. Instead we compute the background from the area surrounding the target region. This makes it even more important to label the approaches. In addition to the DAM method we will refer to the second method as TmS (Target minus Surrounding) in the revised version of the paper. When the linear regression is applied to the anomalies computed with the second method, we still obtain a large offset. The answer to your questions "Could the offset arising from the linear regression (see (1)) be a consequence of this computation of the anomalies ?" is therefore No (we will show that the numerical values of offset and scaling factor change somewhat but not dramatically). As explained above, the main reason for the large offset are the large biogenic fluxes.

Referee: => (1) and (2) feed the conclusion raised by the authors themselves that their method could have been too simple. This could appear as the main reason for the lack of ability to extract a clear signal from the covid-19 crisis. The authors need to have and share a stronger confidence in the relevance of these computations.

Author's response: Concerning "The authors need to have and share a stronger confidence in the relevance of these computations": Please see our response to your comments (1) and (2) above. Concerning: "This could appear as the main reason for the lack of ability to extract a clear signal from the covid-19 crisis.": We do not think that this is the main reason. As explained in our paper, we think that a main reason is the weakness of the signal (which is only 0.1-0.2 ppm). After submission of our manuscript other publications appeared using different approaches to find out to what extent satellite XCO2 retrievals can provide COVID-19 related CO2 emission reduction information (see Chevallier et al., 2020, Tohjima et al., 2020, and Zeng et al., 2020, now also cited in the revised version of our manuscript). The findings of these studies are consistent with the conclusions drawn in our manuscript. This is not a proof that

the limit has already been achieved via our analysis, but this is a strong indication that a significant information loss due to the simple data-driven analysis method used in our publication is not very likely.

Referee: 3) a) Even though it is mentioned from time to time in the discussion (it often comes too late), the spatial and temporal scale of analysis is not really properly characterized, justified, and discussed while it has critical consequences on the outcomes and conclusions of the study. The introduction provides numbers at scales that are not really defined, probably mixing different scales e.g. at lines 100-102 and l110 p4. Comparisons between the FFCO2 signal "at regional scale" and the noise on individual XCO2 data hardly makes sense (e.g. l103-104 p4 and 358-360 p12). Stating that averaging is needed to decrease the random noise (line 105 p4) is surprising: there is no need to average data over the whole East China and 1 month to ensure the strong reduction of the noise, and appropriate analysis at fine scales should denoise the data much better than simple averaging. The large scale ("regional") XCO2 fields are often dominated by the signal from CO2 natural fluxes and the signal from FFCO2 emissions is generally sought at fine spatial scales. Enlarging the scales of analysis can bury the signal from FFCO2 emissions in that of natural fluxes. One could have negative conclusions with "regional scale" analysis and positive ones from analysis at finer scales as acknowledge a bit late by the last lines of the manuscript. b) Even if East China has a relatively high large scale signal from FFCO2 emissions and even if the analysis are focused on the late fall - early spring period, figure 4 strongly suggests that large patterns from natural fluxes overlap the area of analysis, which could strongly bias the linear regression. Biogenic fluxes and anthropogenic emissions follow a seasonal cycle with a very similar phase and thus the regression may be able to fit the general variations of FFCO2 emissions by assimilating data with a strong signal from the biogenic fluxes. A better care for such a source of uncertainty needs to be taken.

Author's response: Concerning "the spatial and temporal scale of analysis is not really properly characterized, justified, and discussed": We will specify this now better

in the Introduction section where we will write: "Here we use an ensemble of satellite retrievals of XCO2 to determine whether COVID-19 related regional-scale (here ~20002 km2) CO2 emission reductions can be detected and quantified using the current space-based observing system." Furthermore, the coordinates of the East China region is specified in Tab. 1. Why we selected this region is also explained in our paper. The scales mentioned at lines 100-102 is consistent with our definition. The scale relevant for the sentence line 110 is a different one as we here refer to individual satellite ground pixels (footprint). We will improve this sentence to avoid misunderstandings. We will also revise the part where the noise of the satellite data is mentioned. You are right. Noise is not the main reason. The main reason for the selected spatial and temporal resolution is that our method cannot deal with high-spatial and high-temporal XCO2 variations. We use averages to eliminate high-frequency variability. As explained above we will add more information on biogenic fluxes. We will add to the revised Fig. 5 (using the CT2019 data set) XCO2 anomalies due to fossil fuel (FF) emissions and biogenic (BIO) fluxes in addition to the (total) XCO2 anomalies. A comparison of these three different XCO2 anomaly time series permits to identify the contribution of the FF and BIO components to the (total) XCO2 anomaly. We will show that during October to May FF (typically) dominates but the BIO impact is not zero. The non-negligible BIO impact introduces an error, which we quantify.

Referee: 4) The authors claim several times that their method is "data driven" and does not rely on a priori knowledges about the fluxes and atmospheric transport. This is contradicted by the use of the CT2019 system to derive the linear regression between XCO2 spatial anomalies and FFCO2 emissions. This is a critical point: the authors do need to simulate the link between emissions and concentrations using a proxy for the atmospheric transport and some assumptions regarding the spatial distribution of the FF emissions and the other fluxes. Rather than ignoring the atmospheric transport and a priori knowledges on fluxes, they use a crude proxy for it, which could bring large uncertainties to the analysis.

10000

Author's response: We established the link between the XCO2 anomalies and the CO2 FF emissions using CT2019. We use CT2019 to get the two parameters (offset and scaling factor) of the empirical linear relationship used to convert the satellite-derived XCO2 anomalies to FF emissions. Furthermore, we use CT2019 for error analysis. Apart from these two parameters we derive the emissions and their changes entirely from the satellite Level 2 data products without any additional modelling. This is a major difference to more traditional inverse modelling approaches which require very detailed modelling and typically several assumptions, e.g., related to the a priori fluxes and their variances and co-variances. We therefore think that the terms "data driven" is an appropriate characterization of our method.

Referee: 5) In principle, the coarse spatial resolution of CT2019 could be a problem for the analysis of the signal from FFCO2 emissions in XCO2 data which have much finer spatial footprints. How are the XCO2 values derived from this model ? When applying the "DAM" to CT2019 and then deriving the linear regression from this system, is CT2019 subsampled at the satellite locations ? if yes, why would there be one equation only for the linear regression while the authors apply it to four satellite datasets with different samplings ? if not, can the linear regression from full emission and concentration fields really apply to the satellite data, which have a sparse spatial sampling ?

Author's response: No, we do not sub-sample CT2019. For the revised version of the paper we will write this when describing the use of CT2019: "The XCO2 has been computed by vertically integrating the CT2019 CO2 vertical profiles (weighted with the surface pressure normalized pressure change over each layer). The model data are sampled at local noon, which is close to the overpass time of the satellite data sets used here. The spatio-temporal sampling of a specific satellite XCO2 data product is not considered here, i.e., we use the CT2019 data set independent of any satellite data product apart for the sampling at local noon.". As our East China target region is large compared to the resolution of the CT2019 data set, we do not think that the resolution of CT2019 causes a significant problem. Note also that the CT2019 data set

only covers 2015-1018 whereas we use satellite data also in 2019 and 2020. As will be better explained in the revised version, we focus on differences between the October 2019 to May 2020 period and previous October to May periods to find out to what extent the October 2019 to May 2020 period is significantly different compared to the other periods taking into account the standard deviation of these differences. We do this for all satellite products and in the revised version even using a second alternative method (to not only rely on the DAM method). The scatter of the results provides us with error bars considering year to year variability but also shortcomings of our method assuming that our model is not able to capture all variability (which is unlikely the case due to the simplicity of our model). We aim at carefully considering variability and uncertainty when formulating our "careful conclusions" (as written by you in your first paragraph).

Referee: 6) Some parts of the text have been written too quickly (with some lack of clarity, repetitions, copy paste of paragraphs from other publications to describe the model or methods, some typos...). The main text should not detail the color code of the figures (the figures should be clear enough). I think that there are too many figures inserted in the main text. Some could be moved to supplementary material. Others could be merged together into synthetic figures.

Author's response: We replaced all quotes from other publications by appropriate short summaries. We also revised many figures. We think, however, that all figures are relevant and we therefore have not been able to reduce the number of figures. However, we simplified several figures (Figs. 8-12) by slightly modifying our method (the OND anomalies are now computed one stage earlier (Fig. 7)). As a consequence, most of the numerical values listed in Tab. 4 also changed slightly. The overall conclusions are however not affected.

Referee's Secondary points:

Referee: - the authors indicate that this study is the "first attempt to determine whether a regional-scale reduction of anthropogenic CO2 emissions during the COVID-19 pandemic can be detected using space-based observations of atmospheric CO2" (p1 l20) -> given the speed of some scientific studies and the rapid publication of papers, especially during the covid-19 crisis, this claim may not hold. It may depend on the precise definition for the term "regional scale" used here. In any case, such a fact should not lower the need for strong justifications for the simple and quick analysis methods applied here (see (1) and (2)).

Author's response: In the meantime, additional publications have been published. We added references to these publications to our manuscript and revised the "first attempt" statement (we now plan to write: "Our study is one of the first attempts . . .").

Referee: - a source of misunderstanding when discussing increase of decrease of emissions due to covid-19 is whether comparisons apply to previous months or to identical calendar month of the previous year(s). Things are even more complex here where changes from year-1 to current year of changes from month-1 to current month of emissions are analyzed. Even though it may hamper the fluence of the text, the description of the anomalies should systematically be clear regarding this if the context of the sentence does not help (e.g. line 42 on p2, line 83 p3, line 352 p12).

Author's response: For the revised version of our manuscript we will aim at improving our manuscript also with respect to this aspect.

Referee: - Should we understand from this paper that the anomaly of FFCO2 emissions due to covid-19 can be quantified from the TCCON network (line 94 p3) but not from the XCO2 GOSAT and OCO-2 data even though the TCCON network is too sparse to assess the satellite data accuracy at regional scale in East China (end of introduction) ?

Author's response: No. This is a misunderstanding. We will revise our manuscript to avoid this.

Referee: - The "complementarity" between OCO-2 and GOSAT (line 126 p4) could be

further discussed. So far, I feel that there has not been much expectations regarding the ability of GOSAT to allow for the quantification of FFCO2 emissions, while various studies have attempted at quantifying FFCO2 emissions based on the OCO-2 data.

Author's response: There are also several publications using GOSAT data for this purpose, see, for example, Matsunaga and Maksyuto, 2018, cited in our manuscript.

Referee: - l162 to 169 p5-6 display a loose discussion on whether the width of latitudinal bands is important or not, with some opposition between l163 vs. l162 and 168-169. The "good agreement" discussed on line 169 sounds qualitative while the anomalies will be used for the quantification of emission temporal variations in East China. Some quantitative assessment of the sensitivity to the width of latitudinal band would have made more sense.

Author's response: We will replace "good agreement" with a more appropriate term. To address these comments (and the related comment listed above) we have now also used a second method not depending on latitude bands and will also report results obtained with this alternative method in the revised version of our manuscript.

Referee: - isn't the notation "deltaXCO2FF" for the estimate of the emissions misleading ? this is an estimate of the total FF emissions, not of the FF component of the XCO2 anomalies.

Author's response: Agreed. For the revised version of the manuscript we will use a different notation.

Referee: - table 1: should not the labels "generated by the authors (#)" be changed ? table 3 is useless, the coordinates should just be given in the text.

Author's response: Agreed. We will modify the table along the lines suggested.

References:

Chevallier, F., Zheng, B., Broquet, G., Ciais, P., Liu, Z., Davis, S. J., et al. Local anomalies in the column-averaged dry air mole fractions of carbon dioxide across the globe during the first months of the coronavirus recession. Geophysical Research Letters, 47, e2020GL090244, https://doi.org/10.1029/2020GL090244, 2020.

Hakkarainen, J., Ialongo, I., Maksyutov, S., and Crisp, D.: Analysis of Four Years of Global XCO2 Anomalies as Seen by Orbiting Carbon Observatory-2, Remote Sensing, 11, 850, doi:10.3390/rs11070850, pp. 20, 2019.

Matsunaga, T., and Maksyutov, S. (eds.): A Guidebook on the Use of Satellite Greenhouse Gases Observation Data to Evaluate and Improve Greenhouse Gas Emission Inventories, Satellite Observation Center, National Institute for Environmental Studies, Japan, 1st Edition, March 2018, pp. 137, https://www.nies.go.jp/soc/doc/GHG_Satellite_Guidebook_1st_12d.pdf (last access: 26-Aug-2020), 2018.

Tohjima, Y., Patra, P.K., Niwa, Y. et al. Detection of fossil-fuel CO2 plummet in China due to COVID-19 by observation at Hateruma. Sci Rep 10, 18688. https://doi.org/10.1038/s41598-020-75763-6, 2020. Zeng N., Han P., Liu D., Liu Z., Oda T., Martin C., Liu Z., Yao B., Sun W., Wang P., Cai Q., Dickerson R., Maksyutov S. Global to local impacts on atmospheric CO2 caused by COVID-19 lockdown. https://arxiv.org/abs/2010.13025, 2020.

---

## Author Response (AR2)

Bremen, 10-Feb-2021

**Letter to Editor**

for manuscript Buchwitz et al., MS No. amt-2020-386

"Can a regional-scale reduction of atmospheric $CO_2$ during the COVID-19 pandemic be detected from space? A case study for East China using satellite $XCO_2$ retrievals"

Dear Editor,

We have (again) revised our manuscript. We modified our manuscript to take this comment (and only this comment) into account:

Associate Editor Decision: Publish subject to minor revisions (review by editor) (10 Feb 2021) by Ralf Sussmann

Comments to the Author:

Thanks for performing changes as requested by the referees. We have two requests prior to final acceptance.

1. We do not agree to some revisions performed within the 5th paragraph of the Introduction (Author´s tracked changes lines 98-120):

TCCON observations have been the first XCO2 measurements used to explore a COVID-19 impact - published in July 2020. So we would consider it fair, that reference to this TCCON work should be made at the very beginning of this paragraph (as was in the original manuscript) and not at the end of this paragraph as done in the revised manuscript - after referring to various subsequent studies all published after October 2020. I.e., we request to use the original wording for referencing to the previous work by TCCON.

2. Properly use the doi and not a journal-weblink within the reference to TCCON work.

This means that we exactly modified the manuscript as requested by items 1. and 2. This can also be seen in the track changes pdf file which we submit along with the revised manuscript.

We hope that the manuscript is now acceptable for publication in Atmos. Meas. Tech.

Many thanks for acting as Editor for our manuscript! This is very much appreciated!

With kind regards,

Michael Buchwitz

(on behalf of all co-authors)